# A Review on the Design and Hydration Properties of Natural Polymer-Based Hydrogels

**DOI:** 10.3390/ma14051095

**Published:** 2021-02-26

**Authors:** Abdalla H. Karoyo, Lee D. Wilson

**Affiliations:** Department of Chemistry, University of Saskatchewan, 110 Science Place, Saskatoon, SK S7N 5C9, Canada; abdalla.karoyo@usask.ca

**Keywords:** hydrogels, water, hydration, natural polymers, swelling, structure

## Abstract

Hydrogels are hydrophilic 3D networks that are able to ingest large amounts of water or biological fluids, and are potential candidates for biosensors, drug delivery vectors, energy harvester devices, and carriers or matrices for cells in tissue engineering. Natural polymers, e.g., cellulose, chitosan and starch, have excellent properties that afford fabrication of advanced hydrogel materials for biomedical applications: biodegradability, biocompatibility, non-toxicity, hydrophilicity, thermal and chemical stability, and the high capacity for swelling induced by facile synthetic modification, among other physicochemical properties. Hydrogels require variable time to reach an equilibrium swelling due to the variable diffusion rates of water sorption, capillary action, and other modalities. In this study, the nature, transport kinetics, and the role of water in the formation and structural stability of various types of hydrogels comprised of natural polymers are reviewed. Since water is an integral part of hydrogels that constitute a substantive portion of its composition, there is a need to obtain an improved understanding of the role of hydration in the structure, degree of swelling and the mechanical stability of such biomaterial hydrogels. The capacity of the polymer chains to swell in an aqueous solvent can be expressed by the rubber elasticity theory and other thermodynamic contributions; whereas the rate of water diffusion can be driven either by concentration gradient or chemical potential. An overview of fabrication strategies for various types of hydrogels is presented as well as their responsiveness to external stimuli, along with their potential utility in diverse and novel applications. This review aims to shed light on the role of hydration to the structure and function of hydrogels. In turn, this review will further contribute to the development of advanced materials, such as “injectable hydrogels” and super-adsorbents for applications in the field of environmental science and biomedicine.

## 1. Introduction

Hydrogels are polymeric materials capable of swelling and maintaining a distinct 3D structure upon absorption of large amounts of water. The first synthetic hydrogels were developed in 1954 by Wichterle and Lim [1] using poly-2-hydroxyethyl methacrylate (pHEMA). Soon after this discovery, hydrogels were used in contact lens production [1,2,3] and have since been used in various applications, including tissue engineering, drug delivery systems, environmental remediation [4,5], agriculture and biosensors [6,7,8,9,10]. Traditionally, biomaterial hydrogels were designed mainly by crosslinking polymerization reactions. However, recent advances in the chemistry of biomaterials has revitalized this field of research. The physicochemical, mechanical, and biocompatible properties of hydrogels depend on the polymer type, constituent ratio, composition, and the fabrication process. Various novel design approaches were reported to afford versatile hydrogel materials with enhanced mechanical properties [3,4,5,6], super-absorptivity [11,12], super-porosity [13,14], stability [15], and multi-responsive character to external stimuli [16]. In particular, the water sorption property and stability of hydrogels relate to a great number of biomedical and technological applications such as drug delivery systems, soft contact lenses, enzyme immobilization, artificial implants, and flocculants for the removal of heavy metals [17]. A number of classifications of hydrogels are reported in the literature as shown in Figure 1 accounts for a diversity of materials with variable physicochemical and mechanical properties [6,18,19]. Hydrogel systems can be categorized based on their surface/electrical charge as ionic (i.e., anionic or cationic), neutral, amphoteric (containing both acidic and basic groups), or zwitterionic (containing both anionic and cationic groups). Similarly, the type of crosslinking determines whether the hydrogel is physical (e.g., hydrophobic association, chain aggregation, and hydrogen bonding), chemical (e.g., covalent bonding) or has a dual-network of crosslinks [19]. Hydrogels have also been classified based on their structure as either crystalline, amorphous, or semi-crystalline. On the other hand, the source of the polymer scaffold relates to the formation of natural, synthetic, or hybrid hydrogel materials. Hydrogel-forming natural biopolymers include proteins (e.g., collagen and gelatin) and polysaccharides (e.g., starch, cellulose and chitosan). In the case of synthetic hydrogels, various polymer materials including pHEMA, poly(N-isopropyl acrylamide) (PNIPAM), polyacrylamide (PAA), polyvinyl alcohol (PVA), polyethylene glycol (PEG) and their derivatives are well known [7]. Synthetic polymer hydrogels possess high water sorption/retention capacity, longer shelf-life and enhanced gel strength [20]. By contrast, bio-based hydrogels offer added advantages due to their enhanced biocompatibility and biodegradability, as well as good mechanical strength and porous structure imparted by their hierarchical structure [21]. With the advances in materials science, the fabrication of composite hydrogel materials offer advanced hybrid systems with unique physico-mechanical properties. In particular, the synthesis and modification of such systems can be achieved via physical and/or chemical means, where processes such as crosslinking, grafting, impregnation, incorporation, blending, interpenetration, and imprinting methods are involved [8]. The feasibility of applying bio-based hydrogels and their modified forms for biomedical purposes (e.g., tissue engineering, drug delivery systems, and contact lenses), the environment (e.g., flocculants and sorbents for treatment of sludge and wastewater) and agriculture (e.g., release of agrochemicals and water reservoirs) depend on several factors, such as water sorption properties, stability, and mechanical strength.

The design strategy of super-porous and super-sorbent hydrogel materials for application in the environment and the biomedical fields demands the formation of highly functional 3D structures with favorable physicochemical and mechanical properties. Problems such as low solubility, high crystallinity, non-biodegradability, low gel strength, and unfavorable mechanical and thermal properties limit the fabrication and/or application of hydrogel materials. Other limitations may arise if the fabrication processes are associated with unreacted monomers and the use of toxic crosslinkers. Advanced hydrogel materials with a desirable water sorption/retention capacity and mechanical properties can be achieved via a composite formation strategy that employs natural and synthetic polymers with pre-determined physicochemical properties such as biodegradability, solubility, crystallinity, surface/textural properties, biological activities, and sensitivity to external stimuli. At present, super-sorbent hydrogels consisting of synthetic polymers dominate the market because of their ease of production. However, concerns related to waste disposal, increasing prices of petrochemical feedstock and the desire to use renewable resources are driving the interest to explore sustainable and natural polymers such as starch, chitosan, and cellulose [22]. Super-porous bio-based hydrogels with high water-capacity represent a new generation of materials with increasing demand and interest for advanced technology in fields such as environmental remediation and biomedicine [17]. Thus, these materials can serve as model systems to study the kinetics of water sorption and solute (biopolymer)-solvent (water) hydration phenomena. According to a report by Gibas and Janik [23], the swelling of a hydrogel material is a complex phenomenon that involves various steps of water uptake processes without the dissolution of the polymer network. The hydrophilicity of the polymer network imparted by the presence of polar groups (e.g., –OH, –NH_2_, –COOH, and –SO_3_H) mediates the swelling of biomaterials in aqueous environments. Processes associated with capillary effects and osmotic pressure gradients provide other essential mechanisms of water ingression within the 3D network of the hydrogel material. It is noteworthy that the interconnectivity of the hydrogel network structure maintains the stability of the material, where the rubber elastic forces of the crosslinked junctions counteracts the solubility effect [19,24], as discussed in later sections herein (*vide infra*). Studies on the classification, preparation, and application of hydrogel polymer systems based on natural, synthetic, and hybrid systems are widespread [6,13,14,17,18,19]. However, studies related to the swelling behavior and kinetics of water absorption for such materials are limited [22]. In turn, this study aims to compare the swelling and gelation behavior of structurally similar natural polymers such as starch, cellulose, and chitosan in an effort to address knowledge gaps related to their structure-sorption property relationships and to catalyze further studies related to such biopolymer hydrogel systems.

In this review, we report on the nature and role of water in the formation and stability of various types of hydrogels based on natural polymers and hybrid systems. This review outlines different types of responsive hydrogels and their various design strategies. The use of natural polymers (e.g., alginate, collagen, starch, cellulose, and chitosan) in the fabrication of hydrogel materials has advanced the field of functional hydrogel systems. This is related to the role of such biopolymer scaffolds to afford the formation of interpenetrating polymer networks (IPNs) with improved water diffusion/sorption and rheological properties [25,26,27]. Since water is an integral part of hydrogels that constitute a major fraction (>90%) of its overall composition, there is a need for an improved understanding of its role governing the structure of biomaterials in relation to the degree of swelling, rheological properties, and the stability of the gel structure. Herein, we cover the interaction of water with the polymer network, whereas various factors that affect hydrogel swelling are highlighted to gain further insight on the role of hydration phenomena. The effect of factors such as the degree of crosslinking, presence of cationic surfactants, and ionic strength of the aqueous medium provide a molecular-level understanding of hydrogel structure. The water sorption properties and the thermal/mechanical stability of hydrogels account for a great number of technological applications, ranging from biomedicine to pharmaceuticals and the environment. Thus, an improved understanding of the processes that govern water sorption phenomena is important for advancing the field of biomaterials by addressing the demand for functional sustainable materials and knowledge of their structure-function properties. The review will contribute to the design of advanced hydrogel materials with improved stability and mechanical strength, such as “injectable hydrogels” and super-sorbents, for applications in the environment and the biomedical fields.

## 2. Fabrication of Bio-Based Hydrogels

The widespread use of hydrogels in biotech applications (e.g., drug delivery systems, tissue engineering, food technology, cosmetics, and environmental remediation) demand that the fabrication process affords materials with the required physico-mechanical properties. Thus, the desired properties of the prepared polymer hydrogels relate to various factors: origin (source) of the biomaterial, its feed ratio/concentration, and the fabrication process by which the hydrogel material is generated. The fabrication of hydrogels can be achieved through dissolution of a single component biopolymer in a solvent to form a 3D hydrogel structure. Similarly, composite systems comprised of two or more polymer units that are crosslinked in a single- or multi-step synthesis to form hydrogels represents a common approach. The 3D crosslinks in hydrogel materials serve to maintain the overall integrity of the hydrogel structure and its ability to swell in water. The interactive features responsible for the water uptake process include capillary, osmotic, and hydration forces that are counter-balanced by various intermolecular forces (e.g., polymer-polymer, solvent-polymer, and solvent-solvent interactions) exerted by the crosslinked polymer chains that limit expansion [28]. Thus, the presence of polar groups in the polymer units (e.g., –NH_2_, –COOH, –OH, –CONH_2_, –CONH, and –SO_3_H) impart a hydrophilic character to the polymer network that favours hydration. On the other hand, the chemical and physical crosslinked junctions maintain the 3D structure of the hydrogel in the swollen state. Whereas the absence of such crosslinking points results in the dissolution of the hydrophilic linear polymer chains due to their thermodynamic compatibility in aqueous media [19]. While fabrication strategies of hydrogel materials that employ single component polymers are rare, the use of multi-component systems that incorporate crosslinker and filler agents, such as low molecular weight bifunctional polymers, bionanomaterials, and inorganic nanoparticles is more common and affords advanced materials with unique and improved properties [20,29,30]. Natural polymers with abundant functional groups and structural diversity enable synthetic modification via grafting, crosslinking, and simple blending processes. Herein, we present some selected examples of hydrogel materials fabricated from single component polymers and composite materials via crosslinking techniques.

### 2.1. Single-Component Hydrogels

In the present work, single-component hydrogels refer to hydrogel systems that are fabricated from a single polymer without the use of crosslinker agents. Many natural polysaccharides (e.g., starch, cellulose, chitosan, agarose, and alginate) and proteins (e.g., collagen and gelatin) extracted from biomass fall into this category of polymer hydrogels because of their unique properties, e.g., high molecular weight, porous 3D structure, hydrophilicity, and biocompatibility. The utility of this category of biopolymer hydrogels is limited due to the lack of elaborate chemical modification to allow for precise control of various properties (e.g., gelation time, gel pore size, chemical functionalization, and degradation/dissolution) that are necessary for their performance for in vivo applications. Single-component hydrogels of chitosan in organic acids (e.g., acetic and lactic acids) have been reported. For example, Qu et al. [31] have synthesized a pH-sensitive hydrogel by dissolving an aqueous chitosan dispersion in lactic acid. The unreacted –NH_2_ groups of chitosan for such hydrogels are responsible for the material’s response to environmental pH variation. More recently, hydrogels of chitosan [32] and starch [33] were prepared by dissolution/melt-blending in carboxylic acids (i.e., acetic and citric acids) for applications in 3D bio-printing. Formation of single-component starch-based hydrogels are known to involve gelatinization, followed by a retrogradation process to form a 3D network [34]. Thus, it is required that the crystalline structure of starch is disrupted by solubilization and/or heating in water or by chemical means using alkaline solution in order to facilitate polymer chain–chain interactions [34,35]. Single-component super-sorbent starch-based hydrogels with high affinity for water and saline were also reported for use in agricultural applications. The hydrogels were prepared by gelatinization of a sulfonate-starch derivative in water at 78 °C and subsequent addition of acetonitrile and initiator chemicals to the cooled slurry (50 °C) to drive the graft polymerization process [36]. Hydrogel systems of native starch in water and ethanol are also known with reported swelling capacity values exceeding 400% [37]. Most recently, hydrogel systems based on ozonated cassava starch were achieved with higher functionality and improved gel strength [38]. Ozone-modified natural hydrogels are considered as an emerging technology that is environmentally friendly with new applications in 3D printing and food. Collagens, particularly type I collagens, comprise ca. 90% of the protein in human connective tissues that have also been used to prepare single-component hydrogels [39,40]. Collagen-fibrils can undergo self-assembly at a neutral pH into bundled fibers that can crosslink to produce matrices that ultimately form hydrogels in the presence of a water-based solvent system.

### 2.2. Multi-Component Composite Hydrogels

Multi-component composite hydrogel systems consisting of two or more polymers are more commonly reported in the literature. These systems are prepared with the use of crosslinkers and/or filler agents to obtain hydrogels with improved physico-mechanical properties. Hydrogel composites can either be formed by physical (e.g., hydrogen bonding, hydrophobic interactions etc.) or chemical (e.g., covalent) crosslinking. The classification of hydrogels as physical or chemical is based on the mechanism of crosslinking, where the process can occur in two scenarios: in vitro during the preparation of the hydrogel, or in vivo (in situ) after application to the target site of interest. The latter has utility in biomedical fields, such as dentistry, orthopedics, and transdermal drug delivery [5,23].

#### 2.2.1. Physical or Reversible Hydrogels

Physical hydrogels are obtained through physical crosslinks that include chain entanglement, hydrogen bonding, hydrophobic interactions, ionic interactions, and crystallite formation [35,41,42,43]. These physical bonds may not be permanent in nature; however, they provide sufficient strength to render the materials insoluble in aqueous media with the propensity to ingress large amounts of water. Physical hydrogels can readily undergo sol → gel transitions in response to changes in external stimuli and have wide utility in environmental and biomedical applications. Formation of hydrogel composites can be achieved by combining a natural and a synthetic polymer (hybrid), or a natural and/or synthetic polymer(s) that include additives such as inorganic nanoparticles where various synthetic methods such as physical blending, graft polymerization, host-guest inclusion, and freeze-thawing are applied. Hennink et al. [43] provided a comprehensive review of various synthetic methods for the design of physically crosslinked hydrogels using amphiphilic block polymers by employing hydrogen bonding, ionic interactions, grafting, and crystallization processes. Biopolymers (e.g., starch, cellulose, and chitosan) and synthetic polymers (e.g., pHEMA, PVA, PEG, PAA) have been utilized due to their diverse chemical functionality and suitable pharmaceutical/biomedical characteristics [41,43,44,45]. Wu et al. have listed various chitosan-based composite hydrogels blended with polymers such as alginate and PEG with potential utility as drug delivery vectors (see Table 1 in Reference [44]).

Hydrogels formed via ionic crosslinking with metal cations (M^n+^), such as Ca^2+^, Fe^3+^, Sr^2+^, and Ba^2+^ represent simple examples of multi-component physical hydrogels, as depicted in Figure 2 [46,47,48,49]. Alginate has been widely reported as a hydrogel precursor due to its gel-forming and M^n+^ chelating ability owing to the large number of COO^−^ groups (cf. Figure 3) [50]. Moreover, alginate has variable binding affinity with different multivalent metals ions that give rise to 3D gel networks with measurably different properties. Narayan et al. [51] have reported photoresponsive Fe^3+^-crosslinked alginate hydrogels for drug delivery systems and tissue engineering. Cation-imprinted alginate beads were reported by Min and Hering [49] for removal of anionic contaminants, such as As, Cr(IV), and Se. These Ca-Fe hydrogel beads were formed by equilibrating alginate solution in Ca^2+^, followed by washing and doping of the alginate-Ca^2+^ composites with Fe^3+^, to allow for partial substitution of the Ca^2+^ ions with Fe^3+^. The latter provides favorable active sites for As(V) to sorb with ≈94% removal efficiency, as reported by Min and Hering [49]. Similar to the case of single component hydrogels, the structural stability and mechanical strength of ionically crosslinked hydrogels depend on various factors that include the source of biopolymer and concentration, along with the degree and density of crosslinking. Moreover, the exchange between monovalent cations from the solvent medium and the multivalent cations from the hydrogel system play an important role in determining the stability of ionically crosslinked hydrogels. For instance, Ca^2+^-crosslinked alginate hydrogel rapidly loses stability (and collapses) in 0.9% NaCl, due to the exchange of Ca^2+^ by non-gelling monovalent Na^+^ ions [52].

Further examples of physical hydrogels employing ionic and hydrogen bonding interactions have been reported. Hussain et al. [54] have designed supramolecular self-healing hydrogels of poly(acrylic acid-co-acrylamide) and hydroxyethyl cellulose (HEC) via hydrogen bonding and ionic crosslinking with Fe^3+^, where the properties of the hydrogel depend on various factors, as described above. The use of cationic surfactants (e.g., dodecyltrimethylammonium bromide; DTAB and cetyltrimethylammonium bromide; CTAB) as non-covalent crosslinkers to form cellulose-starch-based hydrogels was also reported [55]. The cationic surfactants were used to trigger gel formation via electrostatic interactions with the oppositely charged oxidized cellulose nanofibrils (OCNFs), whereas the use of soluble starch served to add higher storage modulus. More recently, Chin and coworkers [56] have reported on citric acid-functionalized starch-based hydrogels prepared by physical crosslinking with PVA and PEG via a freeze-thawing technique. The starch-citrate-based hydrogels exhibited excellent antimicrobial activity against various bacteria strains and sustained release profiles with potential applications as both antibacterial agents and drug delivery systems. Himmelein and coworkers [57] have prepared supramolecular carbohydrate-based hydrogels that combine hydrogen bonding and hydrophobic self-assembly, along with CD vesicles as 3D junctions. In this hydrogel system, HEC was functionalized with hydrophobic adamantyl side groups to form an HEC-AD polymer backbone to act as a guest system. The CD-based bilayer vesicles were self-assembled from synthetic β-CD amphiphiles (β-CDVs) using two types of substituents to act as hosts units; (i) hydrophobic *n*-dodecyl chains on the primary side of the β-CD annular region, and (ii) hydrophilic oligo (ethylene glycol)/PEG units in the secondary annular region. The PEG units face the bulk solvent environment so that the surface of the bilayer vesicles display multiple CD host cavities as shown in Figure 4. The stability and mechanical properties of the supramolecular hydrogel depend on the concentrations of the individual constituents (i.e., HEC-AD and β-CDV) and the preparation temperature conditions. The host-guest inclusion process is the driving force to gel formation, whereas temperature, concentration and host-guest size-fit effects are important factors that contribute to the stability of the hydrogel material. The inability of a smaller α-CD host system to form a hydrogel structure indicates the importance of host-guest interactions in the formation and structural stability of the gel system. Recent examples of supramolecular hydrogel systems based on host-guest inclusion were prepared by Dai et al. [58] using carboxymethyl-cellulose (CMC)-butellinic acid (BA) and α-CD for injectable and thermosensitive drug delivery systems. The host-guest inclusion was achieved between the PEG chains of CMC-BA and α-CD (host) in aqueous environment. Wang et al. [59] have reported novel hydrogel composites by embedding rare-earth compounds (e.g., lanthanide ions NPs) within cellulose hydrogels. These hydrogels have tunable color light emission properties with applications in bioimaging and fluoroimmunoassays. Other recent bio-based hydrogel systems with novel applications are listed in Table 1 (*vide infra*).

#### 2.2.2. Chemical or Permanent Hydrogels

An increased interest in chemically crosslinked hydrogels relate to their favourable mechanical strength. Hydrogels of this type are initiated by covalent crosslinking of polymer units via a crosslinker agent by applying various strategies: (1) reaction of a low molecular weight crosslinker agent (e.g., diacid chlorides, isocyanates, glutaraldehyde, etc.) with a single polymer unit [60]; (2) use of a crosslinked hybrid (polymer-polymer) network (HPN) [61,62], (3) photopolymerization using a photosensitive crosslinker agent [63]; (4) enzyme-catalyzed crosslinking processes [64]; or (5) the use of interpenetrating polymer networks (IPNs), where two or more polymers in the network are formed in such a way that one polymer is crosslinked in the presence of the other [65]. The crosslink points between polymer chains promote 3D network formation that affect the various physicochemical properties of the polymer (e.g., elasticity, viscosity, solubility, and stability) in an incremental way in accordance with the crosslink density and the crystalline nature of the formed hydrogel structure. The 3D network structure along with void imperfections enhance the hydrogel’s ability to ingress large amounts of water via hydrogen bonding.

A variety of natural materials (e.g., starch, chitosan, dextran, and alginate) have been explored for the fabrication of chemical hydrogels for applications in tissue engineering and environmental remediation due to their good biocompatibility, biodegradability, non-toxicity, and excellent gel-forming properties. Recently, Ahmed et al. [66] have reported a comprehensive review on the development of chitin and chitosan-based nanocomposite blends with various organic and inorganic components. These nanocomposites range from hydrogels, microspheres, and films with applications in the biomedical field (e.g., DDS, cancer therapy, and anti-coagulants; cf. Table 1, Reference [66]). Karoyo and Wilson [67] have reported the formation of βCD-based hydrogel by chemical crosslinking with a bifunctional hexamethylene diisocyanate (HDI) molecules. The hydrogel 3D structure was formed via formation of urethane bonds between βCD and HDI. The functional groups of the polymer framework (–NH_2_, –OH, –CO, etc.) and the βCD cavity provided the active sites for hydrogen bonding and hydrophobic interactions with water, respectively. Alginate salts, by virtue of the presence of abundant –COO^−^ groups (cf. Figure 3), form a plethora of crosslinked hydrogel constructs, as outlined in detail elsewhere (cf. Table 1 in Reference [18]). Similarly, starch materials in their pristine and modified forms have good swelling power and mechanical stability with potential utility in various applications [68] in food, pharmaceuticals, cosmetics, sanitary products, cation exchangers, and flocculants, as well as in desiccants for heating, ventilation, and air-conditioning (HVAC) systems [69], including water storage reservoirs for soil amendment [70]. Soluble starch derivatives such as carboxymethyl starch (CMS) and mono-starch phosphates (MSP) have been extensively reviewed. Passauer et al. [68] and Siedel et al. [71] have reported super-sorbent hydrogels using MSP and CMS crosslinked with di- and tri-carboxylic acids (e.g., succinic acid, adipic acid, and citric acid). According to Passauer et al. [68], the free swelling capacity of the phosphate-modified starch-based hydrogels depends on several factors, including the degree of phosphate substitution (DSp), the spacer length of the bifunctional carboxylic acid crosslinker units, and the feed ratio of the crosslinker agent. The degree of phosphate substitution determines the hydrophile-lipophile balance (HLB) of the material, where a greater DSp value results in the collapse of the hydrogel structure due to an increased hydrophilicity (cf. Figure 5a). Similarly, the swelling power of crosslinked hydrogel materials decline when a certain degree of crosslinking is exceeded due to steric hindrance [72,73,74]. Chemical-based cellulose hydrogels were reported using crosslinker agents, such as divinylsulfone [75,76], succinic anhydride [35], and epichlorohydrin [42]. The concentrations of the polymer and crosslinker agent, along with the temperature conditions of the prepared hydrogels, were shown to affect their physico-chemical properties, as described above. Recent examples of bio-based hydrogels are outlined herein (*vide infra*).

## 3. Types and Applications of Stimuli-Responsive Hydrogels

In the preceding sections, general strategies for the fabrication of physical and chemical hydrogels are described that are based on single and multi-component systems. In the case of stimuli-responsible hydrogel materials, more specific design strategies are incorporated to afford materials that undergo sol-gel transitions in response to external stress/stimuli. Stimuli-responsive hydrogels undergo a significant volume phase (or sol-gel phase) transition in response to physical and chemical stimuli. These physical stimuli include temperature, electric and magnetic fields, solvent composition, light intensity, and pressure, while the chemical or biochemical stimuli include pH change, ionic strength, and specific chemical composition. In most cases, conformational transitions are reversible, where hydrogels are capable of returning to their initial state after a reaction as soon as the stimuli are removed. The responsiveness of hydrogels to external stimuli is mainly determined by the nature of the monomer units, charge density, pendant chains, and the degree of crosslinking. The magnitude of the response is also directly proportional to the applied external stimuli [77]. Stimuli-responsive hydrogels are based on a variety of natural (e.g., chitosan, cellulose, and gelatin) and synthetic polymers (e.g., poly-N-isopropylacrylamide, PEG, PAA, PVA, and poloxamers) [78] and have diverse applications in environmental remediation, smart textiles, and drug delivery systems [78,79,80,81,82]. In this section, we briefly outline common types of stimuli-responsive hydrogels with a focus on their design strategy and a description of the key role of hydration in their functional properties.

### 3.1. Temperature Responsive

Temperature is a commonly employed stimuli in hydrogels because of it’s easy to regulate and has relevance to in vitro and in vivo conditions. Thermo-responsive hydrogels have the ability to undergo sol-gel transition as a function of variable temperature. Typically, the structure of these types of hydrogel materials contain hydrophobic and hydrophilic segments, where temperature changes can affect the aggregation between the hydrophobic moieties, and/or interactions with the payload in the case of host-guest systems [82]. Changes in hydrophobic effects and hydrogen-bond interactions may result in variable gel structures accompanied by volume changes at the lower critical solution temperature (LCST). Below the LCST condition, the gel undergoes swelling in a water-based solvent, whereas it shrinks at its LCST. Polymers with lower LCST are typically used as building blocks for thermo-responsive hydrogels. For example, poly(N-isopropylacrylamide) (PNIPAM) is a classic temperature-responsive material used in drug-delivery systems [83]. PNIPAM undergoes a phase transition from a hydrophobic- to a hydrophilic-state in aqueous solution when the external temperature lies below its LCST (ca. 32 °C). This phase transition is triggered mainly by the formation of hydrogen bonds between the amide groups of PNIPAM and water molecules below the LCST. As the surrounding temperature rises above the LCST, the hydrogen bonds between PNIPAM and water are disrupted, and PNIPAM collapses into a globular state, resulting in aggregation of the gel network as it becomes more hydrophobic [84]. Because of their ability to form gels in situ at physiological temperatures, thermo-responsive hydrogels based on natural (e.g., chitosan, cellulose, and collagen) and synthetic polymers (e.g., PEG, PNIPAM, etc.) were reported as sustainable and injectable drug delivery/release agents for various medical conditions such as cancer and central nervous system disorders (e.g., spinal cord and brain injuries) [79,82] (cf. Table 1). The use of “smart textile” applications for transdermal therapy was applied by using drug-loaded thermo-responsive hydrogel excipients [80]. For example, Wang et al. [85] have developed a poloxamer-based thermo-responsive hydrogel to deliver a water soluble traditional Chinese drug to treat an atopical dermatitis disease using smart textiles (cf. Figure 6). The formation of thermo-responsive hydrogel from polymer chains using temperature change as a trigger is schematically represented in Figure 7 [80]. In Figure 7, the release of the drug is triggered at higher temperatures as a result of the collapse of the 3D structure of the hydrogel due to modification of its HLB character and the hydration properties of the material.

### 3.2. pH Responsive Hydrogels

The volume phase transition in pH-responsive hydrogels is determined by the pH variation and the ionic strength of the external environment. These types of materials bear ionic pendant groups (e.g., COO^−^ and –NH) that can accept or donate a proton in response to an environmental pH change. Thus, the degree of ionization (pK_a_) of the hydrogel is dependent on the pH of the medium. Any change in the external pH is accompanied by a net change in the degree of dissociation of the ionizable groups, resulting in an internal and external ion concentration fluctuation. In turn, this perturbation is accompanied by a sudden volume change due to changes in osmotic pressure, and subsequent modified hydration properties driven by electrostatic repulsive forces between ionized groups. Ultimately, the hydrogen-bonded 3D structure of the gel collapses. pH-responsive hydrogels can be controlled for sustained release of drugs at physiological pH conditions. Various natural- and synthetic-polymer based hydrogels have been reported as drug delivery systems for cancer treatment and textile-based transdermal therapy, as described in previous sections. The general mechanism of a stimuli-responsive behavior of pH-responsive hydrogels is presented in Figure 8 [78].

### 3.3. Light Responsive Hydrogels

Light is an interesting stimulus to manipulate because of its ease and convenience of control, both spatially and temporally. Photoresponsive hydrogels may display sol-gel transitions upon irradiation with light and have wide applications in the development of photoresponsive drug carrier systems and materials for tissue engineering [86,87,88]. These materials are typically composed of a polymeric framework bearing a photoreactive functional moiety, such as a photochromic chromophore. Three types of light-induced reactions have been reviewed, whereas some common photosensitive groups have been used [89], such as photoisomerization (e.g., azobenzene, stilbene, and spiropyran), photocleavage (e.g., ortho-nitrobenzyl, triphenylmethane, and azosulfonate), and photodimerization (e.g., coumarin, cinnamylidene acetate, anthracene). In general, the phenomenon of gel-sol transition in such reactions is controlled by various processes, including changes in conformation (e.g., cis-to-trans), bond cleavage, photodegradation of ionizable groups, and formation of dimers. These processes can induce changes in polarity, steric hindrance, and polymer/crosslinking densities to affect the sol-gel transition (desolvated vs swollen or rod-to-coil conformations) of materials, and subsequent release of the guest payload (e.g., drugs and proteins). It is well known that azobenzene-based compounds exhibit typical trans → cis isomerization under UV irradiation and show an unfavorable cis→trans transition under visible light [90,91]. Peng and coworkers [92] have reported a photoresponsive hydrogel composed of trans azobenzene-modified dextran and CD-modified dextran for release of protein (cf. Figure 9). Upon irradiation of the hydrogel system with UV light, the azobenzene moieties isomerize from trans to cis configurations, resulting in the dissociation of the crosslinker units, thereby allowing the entrapped protein to be released into the bulk solution media. Other examples of photoresponsive hydrogels have been reported [93,94].

### 3.4. Multi-Responsive Hydrogels

Multi-responsive hydrogels combining two or more of the properties described above are well documented with diverse functional properties and applications [84,95]. Such materials are characterized by tunable, controllable, and/or biomimetic behavior well-suited for advanced utility in drug delivery and regenerative medicine. Examples of dual-responsive polymer hydrogels featuring pH-temperature, pH-light, and pH-ion responsive properties have been reviewed. More recently, Strachota et al. [83] reported a gel system that contains poly (N-isopropylacrylamide-co-sodium methacrylate) [Poly(NIPAM-co-SM)] intercalated with starch with the capability of rapid swelling-deswelling that undergoes switching in response to temperature and pH. Similarly, a modified form of β-CD was polymerized with N-isopropylacrylamide-itaconic acid (NIPAM-IAM) polymer units to design a dual responsive material for encapsulation and release of drugs in a neutral or alkaline (intestinal) environment [96]. The acrylamide (NIPAM-co-SM and NIPAM-IAM) moieties are responsible for the responsive nature of the hydrogel system to changes in temperature and the pH of the medium. Dual stimuli-responsive supramolecular pseudo-rotaxanes (PRs) hydrogel systems have also been reported that undergo sol-gel transitions due to chemical gradients and pH or temperature fluctuations. Karoyo and Wilson [67] have reported β-CD-based thermo- and chemo-responsive hydrogels with potential utility in environmental remediation and as carrier agents or switchable systems for the controlled release of active ingredients. These hydrogel systems have the ability to encapsulate/adsorb hydrocarbon- and fluorocarbon-based chemicals within the CD cavity sites and the interstitial polymer domains. Similarly, hydrogel materials were reported using α-CD and a star-type molecule (tetrakis-4-hydroxyphenylporphyrin; THPP) end-decorated with benzaldehyde (BA) where the BA moiety can modulate gel formation via pH variation [97]. Recently, more advanced multiple-stimuli responsive hydrogels have been reported by Badeau and coworkers [95], where they designed a series of PEG-based hydrogels with distinct stimuli-responsive crosslinkers, known as “logic gates”. Each of these logic gates is uniquely sensitive to a combination of three environmental triggers (i.e., light, enzyme, or reductant). These hydrogels can respond to specific cues based on a simple Boolean logic principle, “YES”, “AND”, or “OR” (cf. Figure 10). For example, the hydrogels can be instructed to open gates and release their cargo in response to light “OR” reductant, light “AND” enzyme, or light “AND” enzyme “AND” reductant. Other examples of multi-stimuli responsive hydrogels were reported for diselenide-crosslinked starch-based materials [98]. Disulfides and diselenides are widely studied redox-responsive groups whose bonds can be cleaved under specified reducing conditions. The selenide starch-based hydrogels indicated responsiveness to enzyme and oxidative/reductive processes that were tested for controlled-release of Rhodamine B, as a model drug system.

## 4. Structure and Hydration Properties of Biopolymer Materials

The hydration properties of biomaterials, in native and modified forms, represents an expansive subject, where a complete coverage extends beyond the scope of the current review. However, hydration properties represent an important aspect related to the design and utility of hydrogel systems. Such properties are determined mainly by the presence of polar groups and micropore features of the material. Herein, we briefly review the hydration properties of three main biopolymers, namely starch, cellulose, and chitosan. The focus on these biopolymers relates to their similar structures and relevance in the fabrication of hydrogel materials, especially for use in biomedical and pharmaceutical applications (cf. Table 1). Generally, starch, chitosan (chitin), and cellulose are insoluble in aqueous solution at ambient conditions. The molecular structures of these polyglycans are illustrated in Figure 11. The hydration properties of starch and cellulose were reviewed and found to depend on several factors [99], such as the nature of the chemical structure (e.g., HLB character, molecular weight, structural flexibility, porosity, etc.), the propensity of the biopolymer to undergo swelling in water, the accessibility of surface functional groups for hydrogen bonding, and the pK_a_ values. In particular, starch is a major component of many food plants and generally occur as water-insoluble granules. These granules have ordered structures which are semi-crystalline with two main polysaccharides; amylose (AM) and amylopectin (AP). AM is an essentially linear (1→4)-linked α-D glycan, whereas AP is a highly branched polymer consisting of short chains of (1→4)-linked α–D-glucose units with (1→6)-α-linked branches (cf. Figure 11). The functionality of starch is mainly associated with its temperature-dependent interactions with water for well-known processes of gelatinization, pasting and gelation [100,101,102]. When starch granules are heated in water at a characteristic gelatinization temperature (typically 60–70 °C), the granules swell irreversibly to many times their original size, upon interacting with water via hydrogen bonding [37]. Consequently, the AM chains are preferentially solubilized. During the gelatinization process, as water interacts with the polymer chains, the disruption of polymer-polymer hydrogen bonding emanates, with the eventual irreversible loss of crystallite structure of the starch. Pasting refers to the changes in viscosity just before, during, and after the event of gelatinization. Upon cooling of sufficiently concentrated starch dispersions to room temperature, the starch polymer-water hydrogen bonds are replaced by polymer-polymer hydrogen bonds, leading to the formation of a gel network. The polymer-polymer re-association of starch is analogous to the return to its granular state, where this process has been termed as retrogradation [103]. 3D hydrogel starch constructs were prepared to capitalize on their gelling and water uptake/swelling properties, whereas various types of crosslinker agents and polymerization techniques have been applied (vide supra) as shown in Table 1 (*vide infra*).

Cellulose is a high molecular weight polymer consisting of β-1,4-D-glucopyranose units and is the most abundant of nature’s biomass. Cellulose is insoluble in most solvents including water, due to its highly developed inter- and intra-molecular hydrogen bonding between the abundant –OH groups. These intra- and inter-molecular hydrogen bonds cause a parallel arrangement of multiple cellulose chains to form well-defined microfibril aggregates with alternating crystalline and amorphous regions. The chemical structure of cellulose is responsible for its crystallinity and rigidity, along with its water insolubility. However, cellulose exhibits excellent properties (e.g., mechanical strength, biodegradability, hydrophilicity, and biocompatibility) that are responsible for its extensive use in the fabrication of hydrogel materials with diverse applications in biomedicine and pharmaceuticals. Many kinds of chemically-modified cellulose, such as methyl-, hydroxypropyl-, hydroxyethyl-, and hydroxymethyl-cellulose, exist that are water soluble [104]. Cellulose-based hydrogels have been formed, utilizing native cellulose (and its derivatives) in homogenous media (e.g., LiCl and ionic liquids) to solubilize cellulose, and in heterogenous media where cellulose fibers are used to modify the mechanical properties of hydrogels [28].

Chitosan is a cationic polysaccharide derived from chitin, which is the second most abundant renewable polymer after cellulose. It is a linear copolymer composed of randomly distributed β-(1,4)-linked D-glucosamine (deacetylated units) and N-acetyl-D-glucosamine (acetylated units) (cf. Figure 11). Chitosan may possess variable levels of deacetylation since it is derived from chitin (2-acetamido-2-deoxy-β-D-glucan) upon alkaline deacetylation or via enzyme-assisted hydrolysis. Chitosan has a large number of active amino (–NH_2_) and hydroxyl (–OH) groups, where its hydration properties are limited at a neutral and alkaline pH due to its low water solubility and pK_a_ value (ca. pK_a_ 6.2–7.0). The lower pK_a_ value for chitosan indicates that it can be protonated at slightly acidic pH conditions to yield a polycationic polymer with a high density of –NH_3_^+^ groups with improved aqueous solubility [105]. As such, acidic solutions (e.g., acetic acid, citric acid, HCl, and HNO_3_, etc.) are required to prepare solubilized chitosan. Functionalization of chitosan with polar or ionizable groups at the –OH or –NHR (R = H or acetyl) sites yield water-soluble derivatives of chitosan at ambient pH conditions [106,107]. Chitosan continues to attract a great deal of interest for the fabrication of functional materials for biomedical and pharmaceutical applications because of its unique physicochemical properties; biocompatibility, biodegradability, low immunogenicity, and non-toxicity. Chitosan-based 3D hydrogel constructs have been reviewed using various polymerization techniques where various polymers such as glutaraldehyde, PEG, PVA, and epichlorohydrin have been used as crosslinker agents [13,108] (cf. Section 2 and Table 1 in Section 7).

### 4.1. Hydration Phenomena of Biopolymer Gels

Various fabrication strategies were introduced for the design of hydrogels and an overview was provided of the different types of common stimuli-responsive hydrogel systems. Based on the responsive hydrogel systems outlined, it can be inferred that hydration phenomena play an important role in the sol-gel transition of polymer hydrogel systems. The importance of understanding hydration phenomena for a polymer hydrogel relates to its structural composition which constitutes ca. 90% of water. Hydrogels can swell in water or saline up to 1000-fold their dry weight [6], where the amount of sorbed water is usually expressed in terms of equilibrium water content (EWC), as described by Gibas and Janik [23].
(1)$$\;\;\;\;\;EWC=\frac{W_{w}}{W_{t}}\times 100\%$$

In Equation (1), *W_w_* is the weight of the dry gel and *W_t_* is the total weight of the hydrated gel. The significance of water and its interaction with the hydrogel structure lend these materials their unique properties and functions, e.g., modulation of the release profiles of drugs and nutrients. Thus, the function EWC in Equation (1) provides an important characterization of the hydration of hydrogels. Hydration and subsequent swelling of such swellable polymer systems is a multi-step process [23,109], as described here with reference to Figure 12. In the initial step, water molecules hydrate the most polar, hydrophilic active sites of the hydrogel matrix to form the primary bound water (cf. Figure 12). This type of water is an integral part of the hydrogel structure and cannot easily be separated from the hydrogel. In the following step, hydrophobic sites become exposed, interacting with water molecules to form the hydrophobically (or secondary) bound water. The primary and secondary bound waters together form the total bound water with intermediate properties [109]. In the subsequent step, because the osmotic driving force of the polymer network towards infinite dilution is resisted by the covalent or physical crosslinks, an additional amount of water corresponding to the equilibrium swelling capacity of the polymer is adsorbed. The bulk or free water fills the space between the network chains and the centers of larger pores, micropores, or voids [110]. The free interstitial water does not take part in hydrogen bonding, therefore it has properties (e.g., DSC transition temperatures and enthalpy) similar to those of pure water [109].

The total amount of water sorbed by a polymer hydrogel depends on various factors; (i) temperature of the medium; (ii) the nature of the polymer (e.g., surface and textural properties); and (iii) the specific interactions involved between water and the polymer chains. In solution media, the Flory-Huggins theory can be used to describe the adsorbed water, whereas the free energy of mixing (∆G_m_) of the polymer chains and water molecules is described by Equation (2).
(2)$$\Delta G_{m}=\mathrm{kT}\left[n_{w}\ln\phi_{w}+\ln\left(1-\phi_{w}\right)+{\chi n}_{w}\left(1-\phi_{w}\right)\right]$$

Equation (2) shows the dependence of ∆G_m_ to temperature of the medium (T), number of water molecules (n_w_), volume of water (ϕ_w_), and the water-polymer units interaction (χ). The χ parameter depends on the chemical structure and can be calculated from experimental data of the equilibrium water uptake of the polymer network in the presence of a vapour phase containing water [111,112]. The amount and nature of adsorbed water, and the hydration properties of biopolymers, vary according to several factors: hydrophilicity (solubility) of the biomaterial, its crystallinity, along with the structure and textural (surface area and pore volume) properties of the material. For example, the amount of sorbed water varies for starch, cellulose and chitosan due to the differences in their glycan structure and hydration properties, as described above (Section 4).

## 5. The Role of Water and Hydration Kinetics in Biopolymer Gels

The different types of bound water that are essential to the formation of hydrogels were described above; primary-bound, secondary (or hydrophobically)-bound, total-bound and free or bulk water. In general, the role of water in hydrogels has significant implications in their functional properties related to various areas; biomedical fields (e.g., injectable agents, surgical and sanitary pads), agriculture (e.g., water-holding polymers for soil amendment), food industry (food packaging and storage), and construction (e.g., humidity/condensation control). Thus, water is responsible for many of the peculiar characteristics of a hydrogel that affect its 3D structure and properties in terms of stability, rheology and the rate of water (ab)sorption/swelling. Consequently, these characteristics are largely determined by how the polymer network interacts with water, as described above. The role of water and its dynamics in hydrogels can be assessed by monitoring the rates of sorption/desorption, swelling, water retention, and the reproducibility of the hydrogel material (see Section 6). In particular, the mechanism and theories of swelling have been reviewed [113] and provide useful structural information regarding the role and dynamics of water uptake in swellable polymers. Additionally, the influence of various conditions (e.g. salt effect, presence of cationic surfactants, and the degree of crosslinking) on the sorption-desorption kinetics provide further insight on the structure of hydrogels. In this section, an overview of the role of water and its kinetics in natural polymer-based polymers is provided. To further appreciate the role and kinetics of water in hydrogels, various factors that affect water uptake characteristics of the hydrogels will be outlined in later sections (*vide infra*).

### 5.1. Mechanism of Hydrogel Swelling

In order to assess the kinetics of water sorption-desorption processes, it is essential to understand the mechanism by which polymer materials swell in water. The hydration and gelation of starch was described in Section 4.1. Similarly, the fabrication of 3D networks was reviewed for natural polymers (e.g., alginate, starch, cellulose, and chitosan) using such methods as crosslinking and grafting polymerization (Section 2). Crosslinked 3D networks as the one depicted in Figure 12 typically possess abundant hydrophilic (e.g., –OH, –NH_2_, –COO^−^) and hydrophobic functional groups. Water uptake can be mediated via hydrogen bonding interactions at the hydrophilic active sites for both the polymer units and the crosslinker domains. Similarly, water interaction may occur at the hydrophobic sites, as described above. Moreover, the polymer chains are able to stretch due to electrostatic repulsion between polymer units, creating void spaces within the polymer network where a large amount of freely bound water can be absorbed. Hence, it is worth noting that the crosslinking strategy of the polymer during hydrogel fabrication is essential to the formation of a network for various reasons: (i) to increase the number of surface active sites and interstitial domains; (ii) to render the copolymer insoluble in the aqueous environment; and (iii) to enhance the mechanical strength of the polymer network whilst allowing for expansion of the polymer upon water ingression (swelling).

The mechanism of swelling in hydrogels is well described in the literature [113]. In general, the properties of hydrogels for specific applications can be modulated from their bulk properties, including the volume fraction in the swollen state, the corresponding mesh (pore) size, and the molecular weight or spacer distance of the polymer chains between the neighboring crosslink points. The volume fraction describes how much fluid can be sorbed and retained by a polymer. On the other hand, the molecular weight and/or spacer distance between neighboring crosslink points is a parameter that describes the degree of polymerization in a physically- or chemically-crosslinked polymer. In the case of the mesh (pore) size, this parameter represents the space contained within a hydrogel network and is responsible for mediating the diffusion of free water. Hydrogels are commonly classified as either macro-, micro-, or non-porous, in accordance with the sizes of the pores. Jaeger [114] has described the pore size of polymer networks using a structural parameter, the correlation length ξ, which is defined as the linear distance between two neighboring crosslink points. It is noted here that the parameters related to the volume fraction and molecular weight of polymer units can be estimated theoretically or experimentally using the equilibrium swelling and rubber elasticity theories, as described in further detail elsewhere [24,115]. Among the theories, the Flory-Rehner equation [115] is useful in analyzing hydrogels without ionic moieties. In particular, this equation describes the mixing of polymers with water molecules, where two opposing forces can be expressed in terms of Gibbs free energy, as shown in (Equation (3)), where ∆G_elastic_ is the elastic forces in the polymer chains, and ∆G_mixing_ is the result from mixing between the solvent molecules with the polymer chains. This thermodynamic theory states that a crosslinked polymer gel (cf. Figure 12) in solution at equilibrium with its surroundings experiences two opposing forces, the thermodynamic force of mixing and the elastic (retractive) force of polymer chains. The latter contribution favors swelling, while the former hinders swelling [113,116].
(3)$$\Delta G_{\mathrm{total}}=\Delta G_{\mathrm{elastic}}+\Delta G_{\mathrm{mixing}}$$
(4)$${\;\mu }_{1}-\mu_{1,o}=\Delta \mu_{\mathrm{elastic}}+\Delta \mu_{\mathrm{mixing}}$$

The differentiation of Equation (3) with respect to the number of solvent molecules, at constant temperature and pressure gives Equation (4), which describes the chemical potential of the hydrogel. In the equilibrium state, the chemical potential outside the gel is equal to the chemical potential inside the gel (∆μ_1,o_ = ∆μ_1_). That means, the chemical potential from free energy of mixing and elastic forces stored in the stretched polymer chains cancel each other out at equilibrium. In other words, the changes of the chemical potential due to mixing and elastic forces must balance each other [117]. In the case of hydrogels bearing ionic groups, the situation becomes much more complex and an extra contribution (∆G_ionic_) to the total change in the Gibbs free energy is introduced due to the ionic nature of the polymer network.

Whereas the change of chemical potential due to mixing can be expressed using heat and entropy of mixing, the rubber elasticity theory considers a hydrogel as an ensemble of natural rubbers that deform elastically in response to an applied stress. The theory is a modification of previous theories by Teolar [118] and Flory [119] for vulcanized rubbers, and later developed by Silliman [120] and further modified by Peppas and Merill [117] (Equation (5)) for analyzing hydrogel structures prepared in the presence of a solvent.
(5)τ=ϱRTMc(1−McMn)(α−1α2)(υ2,sυ2,x)13

τ is the applied stress to the polymer sample, ϱ is the density of the polymer, R is the universal gas constant, T is the absolute temperature, and M_c_ is the molecular weight between crosslinks. The rubber elasticity theory above can be used to elucidate the structure of hydrogels by analyzing their elastic behaviour, via measurement of such properties as tensile stress. This theory was successfully applied for chemical and physical hydrogel systems [121,122].

It is concluded here that the mechanism of swelling generally encompasses the competing phenomena of polymer-solvent mixing and the elastic stored forces of the stretched polymer network. Thus, efficient hydrogel materials with desirable swelling and mechanical properties require fabrication strategies that account for the abundant surface (–OH, –COO^−^, –NH etc.)/interstitial active sites and flexible 3D network to afford the following: (1) efficient water uptake/retention properties (volume change density), (2) regeneration of the hydrogel material, and (3) enhanced stability/mechanical strength of the hydrogel. The effect of polymer constituents, the degree of crosslinking, surfactant, and salt effect on the water uptake/retention characteristics of 3D polymer networks reveal interesting structural features of such systems (cf. Section 6). In particular, the swelling ratio is related to the charge number and ionic strength of the solution, the nature of the polymer (e.g., the elasticity of the network, the presence of hydrophilic functional groups, and extent of crosslinking density).

### 5.2. Kinetics of Water Sorption

Numerous studies and monographs have reviewed the mathematical modelling of water uptake through diffusion in 3D swellable networks [123,124]. Diffusion-driven water uptake in a substrate (imbibition) is a subject of great interest in the field of food technology, materials science and engineering. As described previously, when a polymer substrate is immersed in water, the water diffuses into the polymer matrix and the material starts to swell. The migration of water by diffusion from the bulk environment into the hydrogel system continues until the process reaches an equilibrium state. The hydration of a polymer can be modelled by two processes; free (concentration-driven), or water demand-driven diffusion. In the latter, the ability of the material to swell becomes the limiting factor in the diffusion process.

Mathematically, the mass transport diffusion for a concentration-driven process is governed by Fick’s second law. The basic form is given by Equation (6):(6)$$\frac{\partial C\left(x,t\right)}{\partial t}=\frac{\partial^{2}C\left(x,t\right)}{\partial x^{2}}$$
C is the concentration, x the distance parameter, t is the time, and D is the diffusion coefficient of water into the polymer matrix. For diffusion into a cylinder and sphere, the parameter x in Equation (6) is replaced by the radial distance (Equation (7)):(7)$$\frac{\partial C}{\partial t}=\frac{1}{r}\frac{\partial }{\partial r}\left(\mathrm{rD}\frac{\partial C}{\partial r}\right)$$
where r is the radius of cylinder or sphere.

The models in Equations (6) and (7) cannot satisfactorily describe a water demand-driven diffusion process, when diffusing molecules cause an extensive swelling of the material such that the ability of the material to swell becomes the limiting factor. An extension of Fick’s law was proposed by Crank [125] and followed up in other studies [22] to compensate for the extensive swelling phenomena of super-sorbent polymers. In general, the mass transport of macromolecular materials (e.g., starch) involves a complex process, as described previously, which can be influenced by many factors. The following parameters play a role: the internal structure of the polymer, glass transition temperature, effects of swelling and relaxation time of the polymer matrix, the chemical nature of the diffusing molecule, and mechanical deformation. Contrary to the proposed extension of Fick’s law described by Crank and other groups, further development of detailed and complex forms of non-Fickian models were proposed, where such models are beyond the scope of the current work. These models describe diffusion of water molecules that are driven by the level of moisture gradients followed by extensive swelling, as observed for starch-based materials. For example, it was estimated that when a starch grain or granule is exposed to water at room temperature, the moisture content of the grain increases only up to ≈0.5 g water/g solid. However, at T_gel_ (60–70 °C), the water swelling capacity of starch is significantly enhanced and contributes immensely to the hydration phenomena [123,124]. The foregoing can be explained from the viewpoint that the water uptake capacity of starch is governed by the temperature-driven gelatinization process, as outlined in Section 3. Witono and co-workers [22] described the diffusion behavior of water into polymer networks in more general terms by taking into consideration the relative rates of diffusion and polymer relaxation. Three basic cases were determined: (I) Fickian diffusion in which the rate of transport is much slower than the relaxation of the polymer chains, where the diffusion of water into the polymer is the limiting factor to the swelling of the polymer; (II) In this case, the diffusion of water is much faster, compared to the relaxation process of the polymer network, where the polymer relaxation process and the restrictions imposed by the network swelling capability of the polymer are the limiting factor to water movement; and (III) The non-Fickian or anomalous diffusion involves an intermediate case where the diffusion of water and the relaxation rates of the polymer network are comparable. A more general and oversimplified model was proposed in the form of an empirical power law to address all of the above situations [22].
(8)$$\frac{M_{t}}{M_{\infty }}=kt^{n}$$

In Equation (8), *M_t_* is the mass of water absorbed at time *t*, *M*_∞_ is the mass of water sorbed at equilibrium, k is a characteristic constant of the polymer, and *n* is a diffusional exponent. The exponent term (*n)* can be obtained from the double logarithmic plot of *M*/*M*_∞_ and *t* and is the key to Equation (8). This is because, its magnitude, which ranges between 0–1.00, determines the type of the water uptake mechanism within the polymer matrix [17]. For instance, for cylindrical hydrogels, *n* = 0.45–0.50 corresponds to a Fickian diffusion process (case I). Whereas, higher *n* values (*n* = 0.50–1.00) indicate non-Fickian or anomalous diffusion (case III), where the dynamic swelling of the polymer and the macromolecular chain relaxations are the dominating factor in the sorption process. At *n* = 0, the mass transfer is independent of time, regardless of the geometry. Ritger and Peppas [126] have demonstrated that the transition from the Fickian to non-Fickian diffusion occurs at a lower value than 0.5 and is also dependent on the geometry of the absorbing material.

## 6. Demonstrating the Water Uptake Characteristics of Hydrogels

In this section, the role of various conditions to the water uptake characteristics of hydrogel materials are outlined. The water uptake behavior of hydrogel polymers can be determined using various parameters, such as water swelling, water retention value (WRV), and reproducibility performance. Further, we demonstrate the effects of such properties as a function of ionic strength and polymer constituents of such hydrogel materials.

### 6.1. Determination of Water Uptake Characteristics

The water absorption (uptake) characteristics of polymer networks can be quantified through measurement of the water sorption/desorption, water retention value (WRV) and swelling rates, as well as the reproducibility performance. The water sorption/swelling rate and reproducibility performance of a hydrogel material are computed using Equations (9)–(11):(9)$$\;\;\;S_{w}\left(\%\right)=\frac{W_{2}-W_{1}}{W_{1}}\times 100\%$$
(10)WAC (gg)=W2−W1W1
(11)$$S_{w}\frac{W_{t}-W_{0}-W_{n}}{W_{0}}$$

In Equations (9) and (10), *S_w_* and WAC are the rates of swelling and (ab)sorption, respectively. *W*_2_ is the weight (g) of the hydrogel after soaking for 24 h (equilibrium swelling) and *W*_1_ is the weight of the dry material after driving off the excess water by filtration or oven drying to constant weight. The water absorption capability has also been calculated by the “tea bag” method using nylon fabric using Equation (11) [75]. In Equation (11), *W_n_* is the weight of the dry nylon cloth, *W*_0_ is the weight of the dry hydrogel before swelling, and *W_t_* is the weight of the swollen hydrogel after immersion in water for time *t* and upon drying by hanging for ≈15 min to remove excess water. Note that Equation (9) can also be used to compute the WRV and reproducibility of the hydrogels where, in this case, *W*_1_ is the weight (g) of the hydrogel upon oven drying (≈50 °C) to a constant weight. In the case of reproducibility measurement, the process of sorption-desorption is repeated several times to determine how the swelling/sorption process can be reproduced over several cycles without the material losing its overall structural integrity.

Similarly, the water desorption rate and retention value provide rich information regarding the dynamics of water in polymer hydrogels. Typically, the moisture desorption rate is determined by allowing a sample of known weight *m* (g) to sorb and desorb water at specified temperature, relative humidity, and time and determined using Equation (12). *D_w_* is the water desorption rate, *m*_1_ and *m*_2_ are the weights of the sample after water sorption and desorption, respectively.
(12)$$D_{w}\left(\%\right)=\frac{m_{1}-m_{2}}{m}\times 100\%$$

### 6.2. Factors Affecting Water Uptake of Polymer Hydrogels

As described above, the water uptake and swelling process of hydrogel composites relate to various chemical and physical forces and the elastic response of the polymer chains. The swelling of polymer networks entail the penetration of water into the hydrophilic matrix by diffusion and capillary action, whereas the elasticity of the crosslink points of the network afford the swelling and contraction of the hydrogel constructs as a result of physical or chemical triggers. Various factors, such as the crosslinking density, specific surface area, the hydrophile-lipophile character and ionic strength of the medium, significantly influence the swelling of hydrogels.

#### 6.2.1. Effect of Ionic Strength (Salt Sensitivity Effect) on Water Absorption

The effect of ions (e.g., K^+^, Na^+^, Ca^2+^ and Al^3+^) on the swelling capability of polymers has been systematically studied [45,127] and provides an important insight into the practical application of hydrogels in vivo at physiological conditions. The phenomenon of swelling-loss has also been reported [127,128] and it relates to the “charge screening effect” of the additional ions (cations or anions). This effect creates a gradient in the mobile ion concentration between the gel structure and the bulk aqueous phase, resulting in osmotic pressure changes and diminished water uptake values. The salt sensitivity factor [$f=1-\left(W_{saline}/W_{water}\right)$], calculated from the ratio of equilibrium swelling capacity in saline solution and deionized water, provides a measure of the effect of ionic strength effects on the swelling property of polymers [127]. The smaller the f-value, the smaller the effect of electrolytes on the swelling capability of the material. Furthermore, the magnitude of the cation charge increases the degree of crosslinking, which may impose a negative impact on the dynamics of swelling.

Zhang et al. and Francisco et al. have investigated the influence of saline solution based on monovalent (e.g., K^+^, Na^+^, and NH_4_^+^), divalent (Ca^2+^), and trivalent (Al^3+^) ions on the swelling properties of chitosan- [127] and starch-based [45] hydrogels. Depending on the cation type (radius and charge), the water uptake for the hydrogel polymers generally decreases in the following order: monovalent > divalent > trivalent cations, where the trend relates to the progressive increase in f values. The higher cationic charge gives rise to greater crosslinking density with correspondingly less swelling. Thus, the relationship between material swelling and cation charge can be summed up using the salt sensitivity factor (f), as described above. By comparison, the effect of f on the swelling capability (g/g) of chitosan-based polymers correlated with the higher magnitude of f as follows: NH_4_^+^ (0.69; 43.3 g/g) < Na^+^ (0.76; 33.6 g/g) < Ca^2+^ (0.93; 10.2 g/g) < Al^3+^ (0.97; 4.7 g/g). Note that multivalent cations were shown to have a higher tendency to be involved in chemical complexation and electrostatic interaction with the functional groups of the polymer matrix to form additional crosslinks [129], as described in Section 2 for ionically-crosslinked hydrogels. Greater crosslink density reduces the degree of swelling due to a reduced accessibility of the interstitial polymer domains. Moreover, an increased ionic strength reduces the difference in the concentration of movable ions between the polymer matrix and the external solution (osmotic swelling pressure) that results in immediate contraction of the gel structure. In a different study by Alam et al. [130], the swelling and WRV of cellulose-based hydrogels in deionized (d) water (725 d-water/g gel) was significantly reduced in saline (s) water (118s-water/g gel) due to charge screening effect (cf. Figure 13).

#### 6.2.2. Effect of Charged Surfactants

The effect of cationic surfactants (e.g., DTAB and CTAB) for the gelation of oxidized cellulose nanofibrils (OCNF)-based biocomposites was briefly mentioned in Section 2 [55]. The effect of charged surfactants (e.g., DTAB and CTAB) on water uptake properties of polymer gels relates to the formation of micelle aggregates, resulting in steric effects and/or depleted active sites for water to bind. For example, the presence of DTAB and CTAB at low concentrations (e.g., 1 mM; Figure 14 (i), (ii) and (v)) was not observed to contribute to the formation of self-standing OCNF gels. However, stable self-standing gels were observed at higher surfactant concentrations, for both DTAB and CTAB (5 mM; cf. Figure 14 (iii) and (vi)). At much higher surfactant concentrations (e.g., 10 mM; (iv) and (vii)), the stability of the gel was greatly reduced, especially for the DTAB system (iv), whereas a significant amount of fibril aggregation led to the loss of optical clarity and concomitant phase separation. The stability of the OCNF/surfactant hydrogels can also be explained by considering the electrostatic interactions between the cationic surfactant headgroups and the anionic cellulose nanofibrils, as indicated by the differences in ξ-potential values between OCNF-DTAB and OCNF-CTAB samples in Figure 14b. Below the CMC values of CTAB (≈1.1 mM) and DTAB (≈14.0 mM), the ξ-potentials of the hydrogel composites remain negative. However, above the CMC of both surfactants, the formation of micelle aggregates is anticipated at the solute (polymer)-solvent (water) interface that results in charge inversion towards the positive ξ-values. Thus, an optimum concentration of surfactant loading is essential to allow for efficient water sorption capabilities of the hydrogels. A similar finding was recently reported in an independent study by Karoyo et al. [131] for composite materials based on starch particles in the presence of a cationic surfactant, cetylpyridinium bromide (CPB). The trend for the observed adsorption was likened to Sabatier’s principle [132], which is commonly ascribed to account for the kinetic processes in enzyme-substrate systems and the interfacial phenomena in biocatalysis. The rise and fall of water uptake properties for the starch-CPB system concurs with a unique surfactant loading, as evidenced by the parallel trend in CPB levels below and above optimum surface coverage values, respectively.

#### 6.2.3. Effect of Crosslinking and Polymer Constituents

The nature and hydrophilicity of a polymer constituent has a direct correlation with the rate of water sorption and retention. In the case of starch-based hydrogels, the difference in the water uptake capacity for different starch materials relates to the relative proportion of the AM and AP polymer constituents [34,133]. The AM content of starch is a greater determinant of starch pasting and gelatinization properties with positive effects in water uptake. For instance, Zhang et al. [45] reported variable sorption properties for different copolymers of starches derived from japonica (JRS-AA; 20%/52%), indica (IRS-AA; 24%/50%), and glutinous (GRS-AA; 1.5%/75%) rice samples, with variable AM/AP percent compositions, as shown in Figure 15. The disparity in the swelling, recyclability, and moisture sorption/desorption rates relates to various factors: (i) the relative composition of the AM/AP within the starches; (ii) the pore structures of the copolymers; and (iii) the propensity of the copolymer networks to swell in water. Generally, starch materials with a higher AM content show greater water uptake properties due to their hydrophilic character, conformational motility, and relative access of donor-acceptor sites [134]. This is further evidenced for the swelling/recyclability rates and water absorption values for the copolymers of IRS-AA with 24%/50% AM/AP contents (cf. Figure 15). However, the inconsistencies in the trends of Figure 15 suggests that the water uptake properties of the copolymers cannot be solely attributed to the AM/AP composition, since other factors (e.g., crosslink density and micropore structure) are likely involved, as indicated by the slow swelling and fast desorption rates for the JRS-AA (20%/52%) sample. Furthermore, an increased AM content above the critical concentration may also result in reduced water uptake and swelling due to extensive crosslinking reactions [17].

Effects of crosslinking were demonstrated in a Na-carboxymethyl cellulose (CMCNa)-hexaethyl cellulose (HEC) composite hydrogel system reported by Astrini and coworkers [75]. This hydrogel was prepared at various CMCNa:HEC ratios (1:1, 3:1, 5:1, and 10:1) via crosslinking with divinylsulfone (DVS). The optimum water absorption and swelling properties of this hydrogel were studied at the 5:1 ratio and 60 °C reaction conditions (cf. Figure 16). Whereas the contribution of polymer components in composite materials is known, the density of crosslinking between the monomer units and the crosslinker agent plays a prominent role in the water sorption for these swellable materials. The greater water uptake and swelling properties for the 5:1 CMCNa:HEC composite at 60 °C indicates optimal crosslinking conditions with accessible active sites for water sorption. Beyond the critical CMCNa concentration (>5 mM), uncontrolled intramolecular crosslinking suppresses the water uptake capability of the polymer network. Moreover, the improved swelling for the polymer prepared at 60 °C corroborates the importance of T in driving the crosslinking reaction.

## 7. Drawbacks and Future Directions of Natural Polymer Hydrogels

Herein, we have outlined the fabrication of hydrogel materials for utility in various applications that range from environmental remediation to foods and the field of biomedicine. In particular, various physical/chemical synthetic strategies (e.g., use of low molecular weight crosslinker agents with a single polymer, crosslinked hybrid polymer networks (HPNs) and interpenetrating polymer networks (IPNs)) to afford functional composite materials with unique properties. Hybrid and/or composite systems offer various advantages to the design strategy of hydrogel systems due to the use of polymer scaffolds with desired properties. For instance, in the case of stimuli-responsive hydrogels, more specific design strategies are incorporated to afford materials that undergo sol-gel transitions under defined external stress/stimuli. The previous sections provide an overview of various examples of stimuli-responsive (pH, temperature, light, etc.) hydrogels and their applications that highlight the fabrication of advanced hydrogel materials with desired properties. The role of water in hydrogel systems was described and constitutes a unique aspect of the structure-function relationship in hydrogel systems and the key importance of hydration properties. In Section 4, the molecular structure and hydration properties of several common types of natural polymers (i.e., starch, cellulose, and chitosan) were described that provide an illustrative overview of the key importance of hydration phenomena in the design of functional composite materials for novel applications. Table 1 outlines a list of recent examples of hydrogel systems that are based on starch, chitosan, and cellulose that highlight a range of novel applications in the biomedical field and the environment.

**Table 1 materials-14-01095-t001:** Hydrogel Fabrication and Novel Applications of Recent (2016–2020) Examples for Systems Based on Starch, Chitosan, and Cellulose.

Systems	Fabrication and Properties	Applications	Reference
**Starch, Acrylic Acid, Organo-Zeolite 4A**	Graft copolymerization of starch, acrylic acid and zeolite. High swelling capability and porosity, salt-responsive swelling-deswelling properties.	Injectable wound dressing with strong antibacterial effects (release of soil nutrients)	(Yan) Zhang et al., 2017 [135]
**Starch, Strontium Ion**	Strontium ion cross-linked starch. Good injectability, good tissue adhesiveness, and strong antibacterial effects.	Injectable wound dressing with strong antibacterial effects	Mao et al., 2018 [60]
**Starch, PVA, Citric Acid, CNFs, Hydroxyapatite**	Starch-PVA crosslinked with citric acid. Highly porous.	Bone tissue engineering	Mirab et al., 2018 [136]
**Starch, Pre-vulcanized Natural Rubber (NR), Sulphur, Glutaraldehyde**	Starch-NR membrane via crosslinking with glutaraldehyde and sulphur used to encapsulate urea beads. Slow release kinetics through the porous membrane (non-Fickian diffusion).	Agriculture and horticulture (release of urea fertilizer)	Vudjung and Saengsuwan, 2018 [137]
**Starch-ozone**		3D-printing applications	Maniglia et al., 2019 [38]
**Starch, PVA**	Starch-PVA crosslinked with glutaraldehyde and blended with a plasticizer (BMIM-BF_4_). High hydrophilicity, plasticity, and porosity.	Oil/water separation	Thakur et al., 2019 [138]
**Starch, Poly(NIPAm-co-sodium Methacrylate) (PNIPAm)**	Copolymerization of PNIPAm in the presence of dispersed starch. Multi-responsive (T, pH), very fast sol-gel transitions, and high swelling/deswelling degrees.	Actuators, solvent- or drug-release systems	Strachota et al., 2019 [83]
**Starch, Citric Acid**	Formation of starch thermoplastic (STP) in molten citric acid and STP-water suspension. Screen printable hydrogel.	Printed electronics applications	Willfahrt et al., 2019 [33]
**Chitosan, Genipin, DHA, L-Carnitine**	Chitosan-acetic acid solution crosslinked with genipin and loaded with skin tanning and anti-aging active components. Slow release kinetics.	Skin tanning and anti-aging	Sole et. Al., 2017 [139]
**Chitosan, Dialdehydes, Diketones**	Chitosan crosslinked with anisole-based phenolic/nonphenolic aromatic dicarbonyls. Affinity for metal ions, good swelling capacity, and biological properties.	Environmental remediation with selectivity for several metal ions and biological applications	Timur and Pasa, 2018 [140]
**Chitosan, Xanthan, Neomycin Sulphate**	-	Topical application, antibacterial agent	Merlusca et al., 2019 [141]
**Chitosan, Acetic Acid, Gelling Agent**	Dissolution of chitosan in acetic acid and addition of a gelling agent. Thermo-sensitive, good cell viability and reduced cytotoxicity.	3D bio-printing agent (cell attachment)	Ku et al., 2020 [32]
**Cellulose, Ianthanide Ions**	Composite blends of cellulose and lanthanide ions. Tunable color light emissions.	Bioimaging and fluoroimmuno-assay	Wang et al., 2017 [59]
**Cellulose, Butelinic acid (BA), α-CD, NPs**	Hydrogels prepared as host-guest inclusion between PEG chains of CMC-BA and α-CD.	Injectable drug delivery	Dai et al., 2017 [58]
**Cellulose, Cylcloextrin (CD)**	Cellulose and CD were crosslinked with epichlorohydrin. Rapid swelling/deswelling kinetics, high loading capacity.	Remotely EMF-induced drug release	Lin et al., 2019 [62]
**Cellulose, Curcumin-Cyclodextrin, Silver NPs**	Microencapsulation of curcumin in HP-β-CD to produce cAgNP which are subsequently loaded into bacterial cellulose hydrogels. Slow-release and antibacterial properties.	Treatment of chronic wounds	Gupta et al., 2020 [142].
**Cellulose, Starch, Amylum, Aluminum Sulfate**	Cellulose and starches were crosslinked using aluminum sulfate. Superadsorbents with high affinity for metal ions.	Environmental remediation with selectivity for heavy metals	Hameed et al., 2020 [143]

While natural polymer hydrogels are attractive due to several properties (e.g., bioavailability, biodegradability, biocompatibility, non-toxicity, and immunogenicity), such materials suffer from their poor mechanical strength. In particular, exposure of natural polymer-based hydrogels to properties such as pH and temperature variations may limit their use in such applications as packaging and in the biomedical fields. Despite their functional diversity, natural polymers are generally insoluble in water due to their complex hierarchical structures, thus presenting a challenge in the fabrication of homogenous bio-based hydrogel materials. The use of weak acids, ionic liquids and/or water-soluble derivatives will continue to contribute to a plethora of advanced hydrogel systems with unique properties and applications. In the case of the poor mechanical strength of natural polymer-based hydrogels, various approaches have been described to address this shortcoming: (*i*) the introduction of crosslinking agents, (*ii*) the use of double network systems, and (*iii*) the use of nanocomposite hydrogels, among other strategies [3]. In the present study, it has been shown that functional hydrogel materials with desirable water uptake/retention capacity and mechanical properties can be achieved via composite formation strategies (*i*–*iii*) using hybrid natural/synthetic polymer units with pre-determined physicochemical properties such as biodegradability, solubility, crystallinity, surface/textural properties, biological activities, and sensitivity to external stimuli. The use of functional crosslinker units, double (or multiple) network systems (e.g., IPNs and HPNs), and the incorporation of nano-sized polymer components particles afford bio-based materials with improved mechanical strength for various application in the environment and the biomedical fields. The incorporation of nano-sized polymer scaffolds in hydrogel systems inhibits the generation of defects and cracks resulting in improved toughening and reinforcement effects. The use of charged polymer depletants, which may fall under strategy (*ii*/*iii*), has been reviewed extensively [144] and contributes to hydrogel structures with improved morphological and mechanical properties. Depletion interactions in macromolecular materials have the potential to cause particle association/aggregation and improve viscosity and rigidity of formed composites, where natural polymers (e.g., polysaccharides) may serve as good depletant candidate materials. Future prospects of natural polymer-based hydrogel systems are bright with design strategies that embed supramolecular systems (e.g., host-guest, stereocomplexation, and biomimetic interactions, etc.) and 3D printing offers new platforms for the development of advanced and unique hydrogel structures. The latter will further contribute to new materials with desirable properties to catalyze the use of sustainable materials to address concerns related to waste disposal challenges and the ever increasing prices of petrochemical-based feedstock.

## 8. Conclusions

This review outlines knowledge gaps related to the role of water and hydration kinetics in various processes for swellable polymer networks. The swelling properties of polymer networks involve the ingression of large amounts water into the polymer matrix via diffusion, adsorption, and capillary action, where the elasticity of the crosslink points of the 3D network afford the swelling and contraction of the hydrogel constructs as a result of physical or chemical triggers. In this review, different types of hydrogels based on single- and multi-component systems are described, along with physical- versus chemical-crosslinked systems. An overview of various types of stimuli-responsive hydrogels provide further insights into the fabrication strategies and structure of stimuli-responsive systems with swelling properties that vary due to the incorporation of polymer units with unique physicochemical features. The role of water and the hydration kinetics of hydrogel systems were accounted for using the Flory-Huggins and the Fickian models. In general, the mechanism of swelling involves competitive processes of polymer-solvent mixing and the elastic stored forces of the stretched polymer network. Hence, efficient fabrication of hydrogel polymer networks should account for abundant surface functional and interstitial active sites, and 3D networks with conformational motility. In general, the Flory-Huggins theory stipulates that the amount of water sorbed by a polymer hydrogel can be modelled as a function of the free energy of mixing of the polymer chains and the amount of water molecules, where factors such as temperature of the medium and the nature of water-polymer interactions govern the hydration capacity of the polymer network. On the other hand, the capacity of the polymer chains to swell in an aqueous solvent is expressed by the rubber elasticity theory and a thermodynamic estimate comprised of two opposing forces (i.e., the force of the solute-solvent mixing and the elastic force of the polymer chains to undergo a volume-change transition). The migration of water from the bulk environment into the polymer 3D network system constitutes an important phenomenon for both the polymer hydration and swelling processes, where the rate of diffusion can be driven either by a concentration gradient or by the water demand (availability). The water uptake capacity of polymer networks can be described by considering the rates of diffusion of water molecules and the polymer relaxation process. Three basic cases were outlined: Case I, Fickian diffusion in which the rate of transport is much slower than the relaxation of the polymer chains. In this case, the diffusion of water into the polymer network is the rate limiting step for the polymer swelling. Case II, the diffusion of water is much faster than the relaxation process of the polymer network, where the polymer relaxation process and the restrictions imposed by the network swelling capability of the polymer is a limiting factor to polymer swelling. Case III, the non-Fickian or anomalous diffusion involves an intermediate case where the diffusion of water and the relaxation rates of the polymer network are comparable. Oversimplified models exist in the form of empirical power laws that encompass all three cases. An account of various parameters, such as water swelling and desorption rates, water retention values, and reproducibility provide valuable structural information related to hydrogel materials. Herein, the effects of polymerization and the surrounding medium (e.g., crosslinking density, salt effect, and presence of cationic surfactants) on the water uptake and swelling capacity of hydrogel systems were determined to provide further insight into the role of water and its hydration kinetics for such systems. The uptake properties of hydrogel polymers are affected by the role of high crosslink density due to steric effects of the polymer units. The salt sensitivity factor [$f=1-\left(W_{saline}/W_{water}\right)$] provides a measure of the effect of ionic strength on the swelling property of polymers. Multivalent cationic species with higher f-values have a greater effect in reducing the water uptake properties of polymers due to their higher chelating power, which may lead to increased crosslinking. Furthermore, the exchange of ions between the solution medium and the hydrogel system can lead to changes in osmotic pressure and subsequent shrinking of the 3D network structure. Cationic surfactants reduce the swelling capabilities of hydrogel systems due to the formation of micelles and/or the reduction of active sites.

## Figures and Tables

**Figure 1 materials-14-01095-f001:**
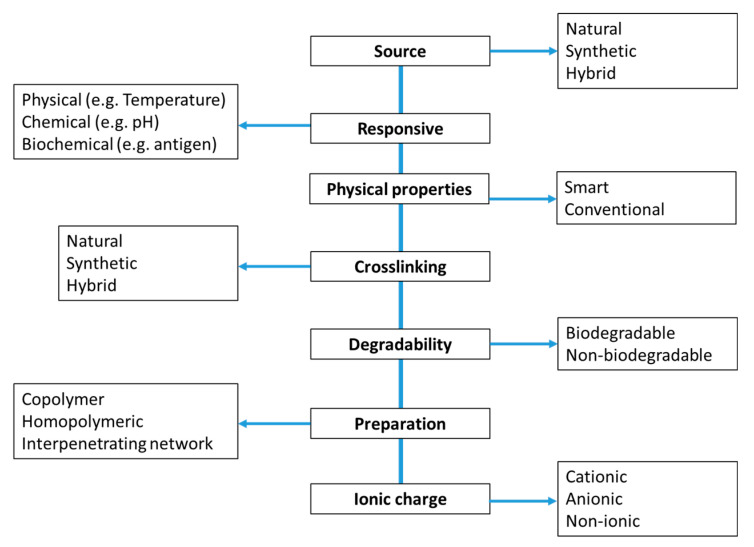
Hydrogel classification based on physicochemical properties. Adapted from Reference [19].

**Figure 2 materials-14-01095-f002:**
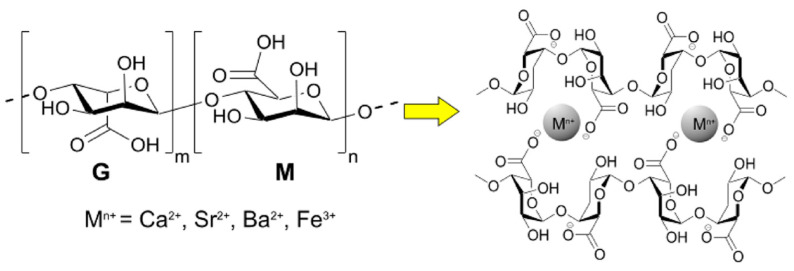
Structure of alginic acid and the ionic crosslinking of alginate by multivalent metal cations (M^n+^). G = α-L-Guluronic acid, and M = β-D-Mannuronic acid residues. Reprinted with permission [51].

**Figure 3 materials-14-01095-f003:**
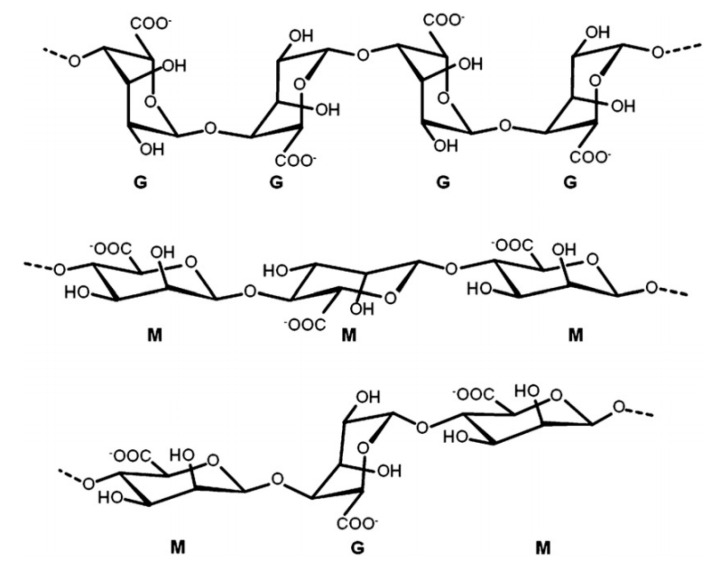
Chemical structures of G-block, M-block, and alternating blocks of alginate, where M = mannuronic and G = guluronic units. Reprinted with permission [53].

**Figure 4 materials-14-01095-f004:**
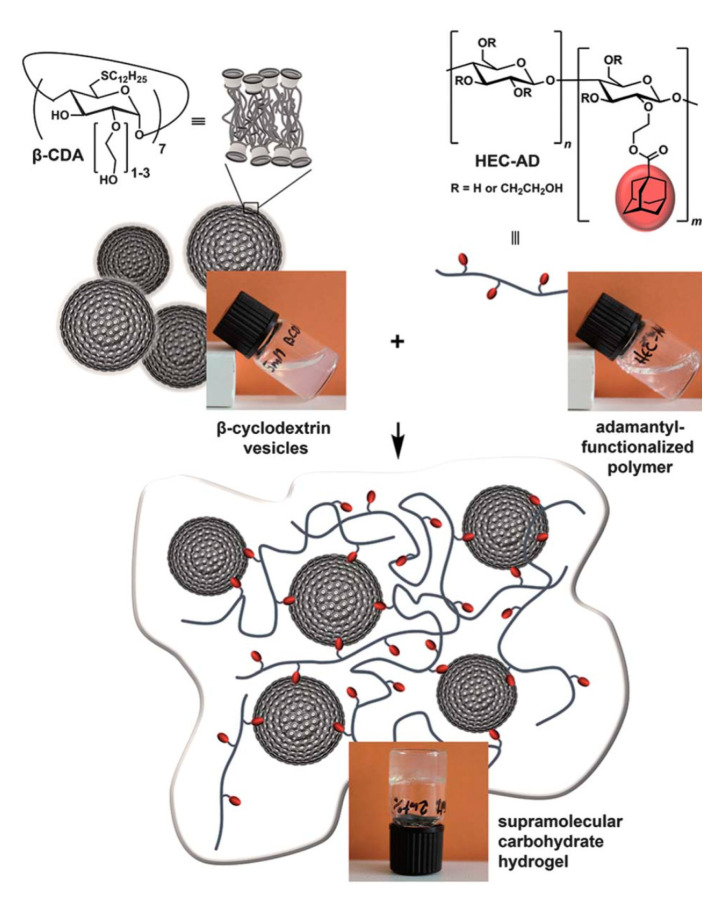
Formation of a supramolecular hydrogel using amphiphilic β-cyclodextrin host (β-CDA) and guest polymer (HEC-AD, m: n ≈ 1:8) and formation of the supramolecular hydrogel by crosslinking of the polymer chains via host-guest inclusion complexes of the adamantane substituents and the macrocyclic hosts on the surface of β-CD vesicles (β-CDV). Reprinted with permission [57].

**Figure 5 materials-14-01095-f005:**
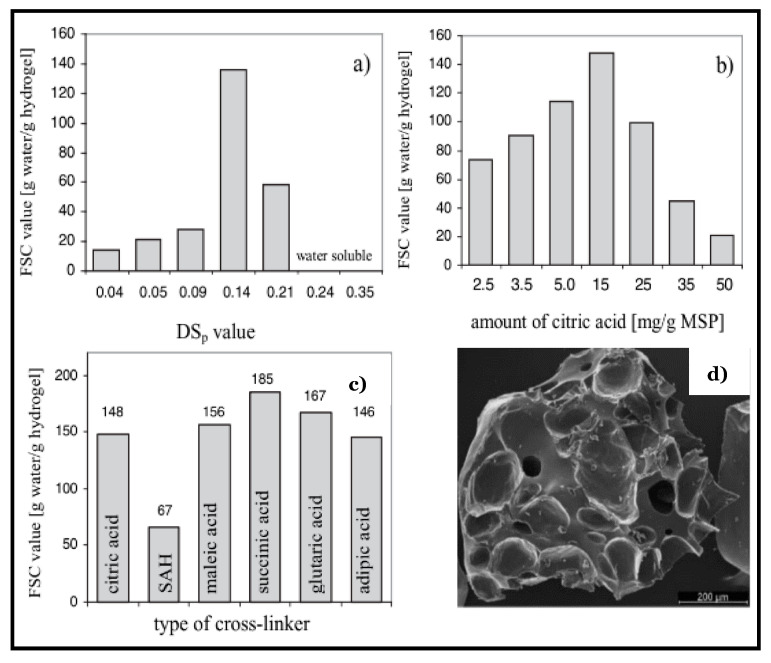
Free swelling capacity (FSC) of carboxylic acid-crosslinked monostarch-phosphate (MSP)-based hydrogels and their dependence on various factors: (**a**) the degree of substitution of phosphate on the MSP, (**b**) citric acid feed ratio, and (**c**) type of the carboxylic acid. Panel (**d**) is an SEM image of the surface of MSP crosslinked with citric acid at a feed ratio of 1:0.015 (*w*/*w*), where the highest FSC value was observed. SAH = succinic acid anhydride. Reprinted with permission [68].

**Figure 6 materials-14-01095-f006:**
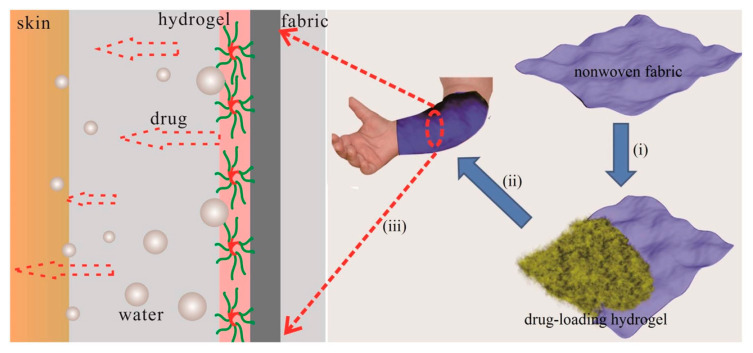
Schematic representation of a thermo-responsive functionalized “smart” fabric hydrogel system with dual-function: (**i**) to coat a drug-loaded hydrogel onto the fabric, and (**ii**) to apply the atopical drug onto the skin of a patient (**iii**) to show how the coated fabric works with moisture and drug diffusion across the skin The drug-loaded hydrogel coated onto the fabric releases the drug, which diffuses across the skin of the patient, as the hydrogel undergoes volume phase transition as a function of moisture variations. Reprinted with permission [85].

**Figure 7 materials-14-01095-f007:**
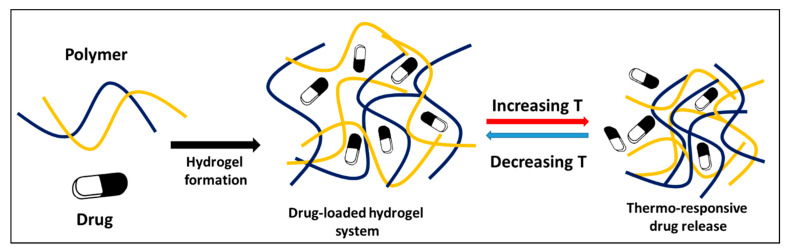
A schematic representation of a load release in thermo-responsive hydrogels upon a trigger by temperature change. The polymer chains form a hydrogel at T < LCST and collapse at T > LCST; thereby releasing the encapsulated drug. Adapted from Reference [80].

**Figure 8 materials-14-01095-f008:**
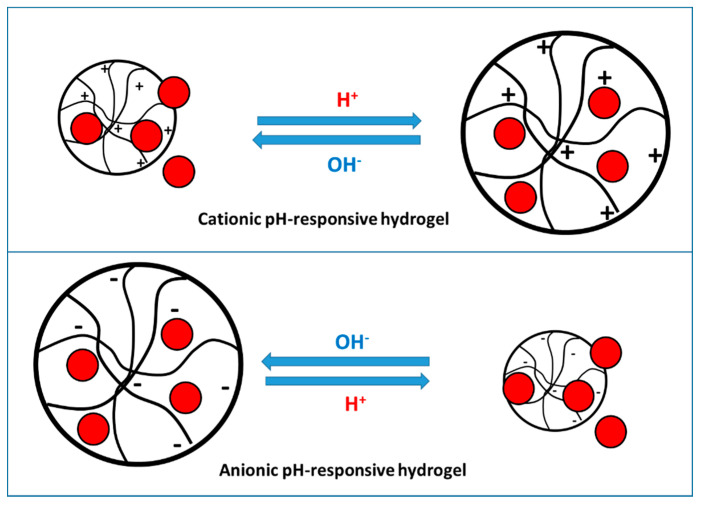
Schematic representation of a general behavior of pH-responsive polymer hydrogel for a drug delivery application. The cationic hydrogel swells at acidic pH and shrinks at basic pH, and releases its cargo. The opposite is true for anionic hydrogels. Adapted from Reference [78].

**Figure 9 materials-14-01095-f009:**
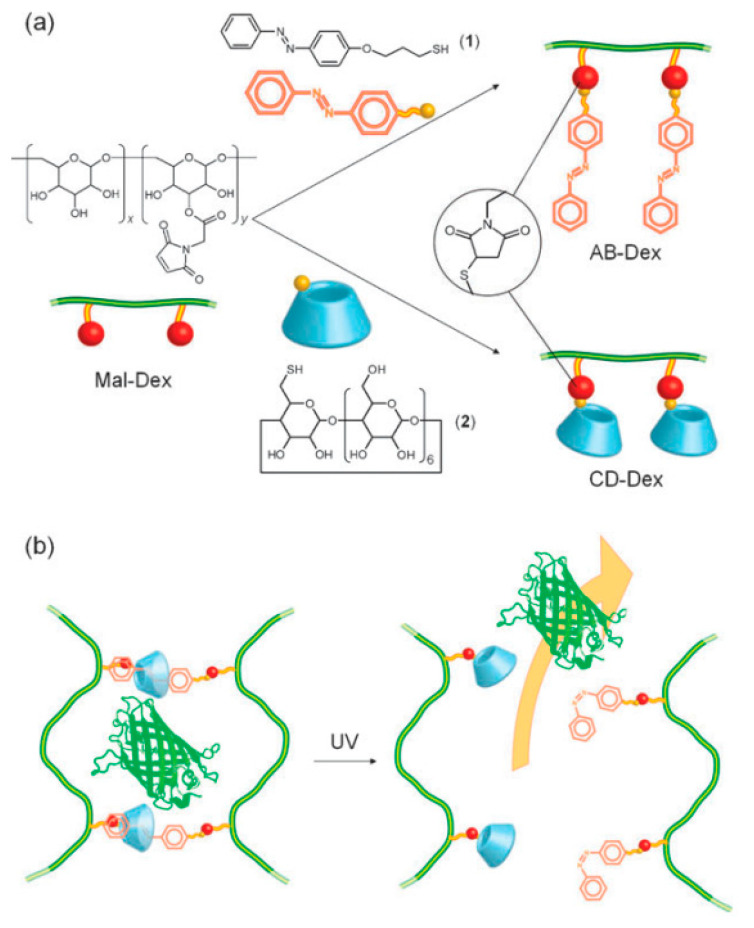
(**a**) Preparation of azobenzene-modified dextran (AB-Dex) and cyclodextrin-modified dextran (CD-Dex). (**b**) Schematic representation of photoresponsive protein release from the trans AB-Dex and CD-Dex hydrogels. Azobenzene moieties isomerize from trans- to cis-forms upon irradiation with UV light, allowing collapse of the hydrogel structure and release of encapsulated proteins. Reprinted with permission [92].

**Figure 10 materials-14-01095-f010:**
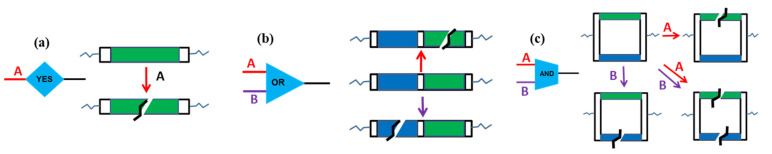
Hydrogels designed with distinct stimuli-responsive crosslinkers (“logic gates”). Chemical crosslinkers enable the hydrogels to respond to specific cues based a simple principle of Boolean logic “YES”, “AND”, or “OR”. (**a**) The YES-gated crosslinker contains a single stimuli-responsive unit (red). (**b**) The OR-gated crosslinker contains two different stimuli-responsive units (red and blue) connected in series. The presence of either relevant input cleaves the crosslinker to allow cargo release. (**c**) The AND-gated crosslinker contains two different stimuli-responsive units (red and blue) connected in parallel. The presence of a single input alone does not fully sever the crosslinker. Adapted from Reference [95].

**Figure 11 materials-14-01095-f011:**
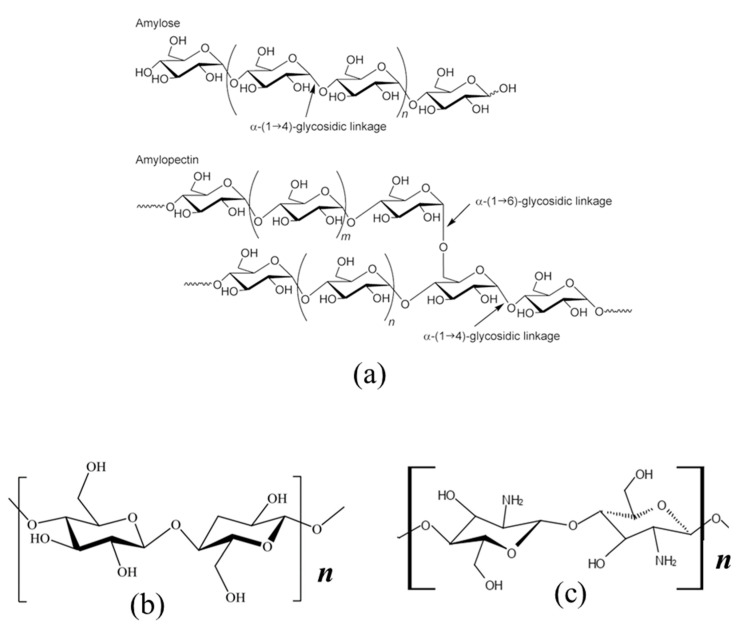
Chemical structures of (**a**) starch, (**b**) cellulose, and (**c**) chitosan (where a –H on –NH_2_ is replaced with an acetyl in the case of chitin).

**Figure 12 materials-14-01095-f012:**
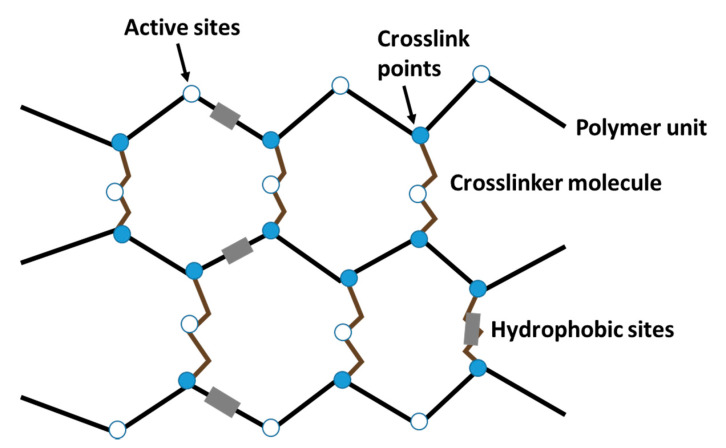
Schematic view of a crosslinked polymer network. The blue dots indicate crosslink joints, white dots indicate polar active sites, and spaces indicate pores or voids. Hydrophobic sites are shown as rectangles. Adapted from Ref. [22].

**Figure 13 materials-14-01095-f013:**
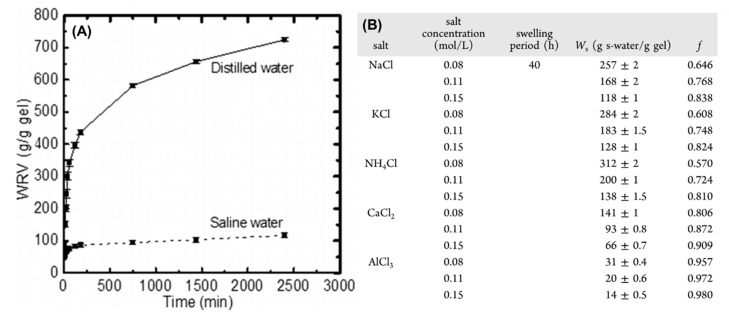
The effect of salt, salt type or salt sensitivity factor (f) on (**A**) WRV; and (**B**) Swelling of CMC-ECH crosslinked hydrogels as a function of time. Reproduced from Alam et al. [130].

**Figure 14 materials-14-01095-f014:**
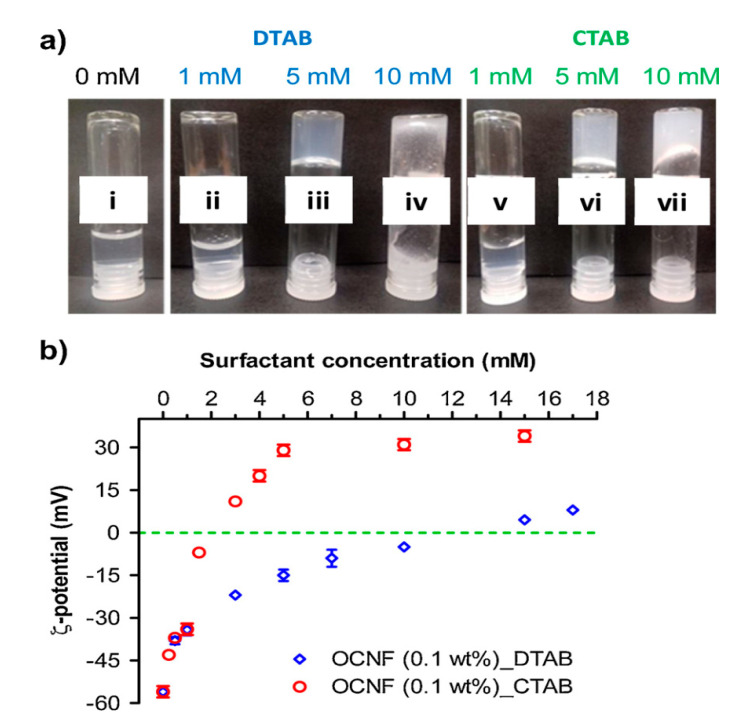
(**a**) Photographic images of cellulose-based (1 wt.%)/surfactant composite hydrogels produced with various loading levels of cationic surfactants: (**i**) Control (0 mM); (**ii**) 1 mM; (**iii**) 5 mM; (**iv**) 10 mM of DTAB; (**v**) 1 mM’ (**vi**) 5 mM and (**vii**) 10 mM of CTAB; and (**b**) ξ-potential values of diluted cellulose (0.1 wt.%)/surfactant systems. Reprinted with permission from [55].

**Figure 15 materials-14-01095-f015:**
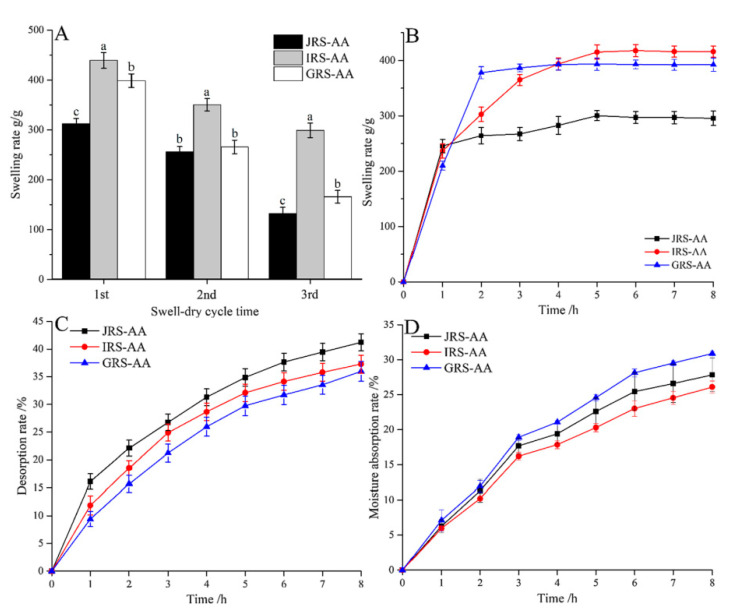
The hydration properties of starch samples with various amylose (%)/amylopectin (%) contents. Japonica (JRS-AA; 20/52), indica (IRS-AA; 24/50), and glutinous (GRS-AA; 1.5/75) rice starch samples showing (**A**) swelling recyclability; (**B**) Swelling rate in water; (**C**) water desorption rate; and (**D**) moisture absorption rate. Figure reproduced with permission from [45].

**Figure 16 materials-14-01095-f016:**
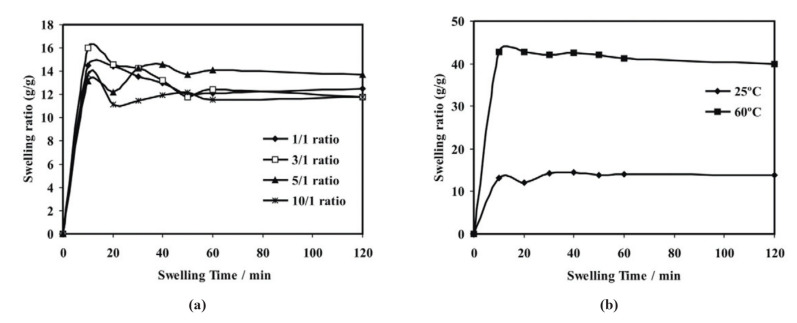
(**a**) Water absorption rate of CMCNa/HEC hydrogels as a function of molar ratio at room temperature; (**b**) effect of reaction temperature on the swelling of the CMCNa/HEC at the 5/1 mole ratio and variable temperature. Reprinted with permission [75].

## Data Availability

The data presented in this study are available on request from corresponding author.

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
