# Peer review of "A Review on the Design and Hydration Properties of Natural Polymer-Based Hydrogels"

_materials, 2021, doi:10.3390/ma14051095_

Round 1
Reviewer 1 Report
The review by Wilson et al concluded current mainstream hydrogels from natural materials. The manuscript covered most of the popular hydrogels in the field, including synthesis, crosslinking mechanism and the specific responsiveness. Overall the manuscript is well structured.
Since the hydrogel field is quite interesting and attractive, there are so many new systems developed beyond the reported ones in this review. I would recommend authors to include more recent studies in this review. This can help to enhance the innovation meaning and more significant in the field. On the other hand, it is recommended to add one more section, specifically about the application examples for the hydrogels.
Based on its current shape, I recommend a minor revision.
Author Response
Reviewer Comments on Hydrogel review for “Materials-1083926”
Reviewer #1 - Comments
The review by Wilson et al concluded current mainstream hydrogels from natural materials. The manuscript covered most of the popular hydrogels in the field, including synthesis, crosslinking mechanism and the specific responsiveness. Overall the manuscript is well structured.
Since the hydrogel field is quite interesting and attractive, there are so many new systems developed beyond the reported ones in this review. I would recommend authors to include more recent studies in this review. This can help to enhance the innovation meaning and more significant in the field. On the other hand, it is recommended to add one more section, specifically about the application examples for the hydrogels.
Based on its current shape, I recommend a minor revision.
Response: The authors agree with the reviewer’s thoughtful comments. Most recent studies covering the last five years, and particularly studies that were published between 2019/2020, were included in the revised manuscript that report recent examples and novel applications of natural polymer hydrogels. A section related to novel applications of natural polymer hydrogels based on starch, chitosan/chitin and cellulose was added in newly added Table (see Table 1) in order to further enhance the innovation meaning and significance of such hydrogel systems in this field.
The authors wish to acknowledge Reviewer #1 for the insightful and constructive comments, along with the opportunity to improve the quality of this manuscript submission.

Reviewer 2 Report
The article is very interesting and bring important knowledge for the field.
The abstract could elucidate the kinetics and the kind of chemical and phisical crosslinkers that are disscussed through out the thext.
Also, if the water can vary achieving 99% in certain gels depending on the polymer, why to show the exactly value of 90%?
About the phisical crosslinkers, the article bellow shows the presence of depletion forces bringing higher strengh with the fibers alignment seen by Crio-TEM
"Effect of depletion forces on the morphological structure of carboxymethyl cellulose and micro/nano cellulose fiber suspensions. JOURNAL OF COLLOID AND INTERFACE SCIENCE, v. 538, p. 228-236, 2018."
Author Response
Reviewer #2 - Comments
The article is very interesting and bring important knowledge for the field.
The abstract could elucidate the kinetics and the kind of chemical and phisical crosslinkers that are disscussed through out the thext.
Response: The abstract was revised to highlight the content of the revised manuscript and to reflect the scope of the review article.
Also, if the water can vary achieving 99% in certain gels depending on the polymer, why to show the exactly value of 90%?
Response: The authors have indicated the composition of water to be approximately 90% (ca. 90%) but it can be much higher. Examples of 400% (or even up to1000%) in the swollen state, as indicated in Section 2.1. The mention of ca. 90% water composition refers to the structure of hydrogel hydration at equilibrium (when its neither in the swollen or de-swollen states).
About the phisical crosslinkers, the article bellow shows the presence of depletion forces bringing higher strengh with the fibers alignment seen by Crio-TEM
"Effect of depletion forces on the morphological structure of carboxymethyl cellulose and micro/nano cellulose fiber suspensions. JOURNAL OF COLLOID AND INTERFACE SCIENCE, v. 538, p. 228-236, 2018."
Response: The specific topic of depletion forces/depletants is now mentioned in the revised manuscript. A general review of the strategies used to address the poor mechanical strength of natural polymer based hydrogels is included in Section7. The use of double network systems and/or nanocomposites may involve depletants/depletion forces as a specific example. We have included some commentary and citations relevant to the depletion forces/depletants in Section 7 as the authors feel that this subject will catalyze more research in the area related to the role of water and its kinetics in bio-hydrogel systems.
The authors wish to acknowledge Reviewer #2 for the insightful and constructive comments, along with the opportunity to improve the quality of this manuscript submission.
Reviewer 3 Report
The paper “materials-1083926-A Review on the Design and Hydration Properties of Natural Polymer-Based Hydrogels” investigates the effect nature and role of water in the formation and stability of various types of hydrogels based on natural polymers and hybrid systems. Here are my questions and comments:
I think it is a parallel work with some new review papers that publish in recent months such as:
“Ali, A., & Ahmed, S. (2018). Recent advances in edible polymer-based hydrogels as a sustainable alternative to conventional polymers. Journal of agricultural and food chemistry, 66(27), 6940-6967.“
or
“Zia, K. M., Tabasum, S., Nasif, M., Sultan, N., Aslam, N., Noreen, A., & Zuber, M. (2017). A review on synthesis, properties and applications of natural polymer based carrageenan blends and composites. International journal of biological macromolecules, 96, 282-301.”
Or
“Bao, Z., Xian, C., Yuan, Q., Liu, G., & Wu, J. (2019). Natural Polymer‐Based Hydrogels with Enhanced Mechanical Performances: Preparation, Structure, and Property. Advanced healthcare materials, 8(17), 1900670.“
so, what makes this review different from the other and from the most recent ones?
Should be provided a comprehensive table between all of the nanofibers until now used.
In Section 6, the Natural Polymer-Based Hydrogels applications should be discussed. What is the future direction of the Natural Polymer-Based Hydrogels use?
Section of drawbacks and future could be increased quality of the manuscript.
A review paper not only should summarize recently published works, but also should contain critical and comprehensive discussions. Therefore, check writing for the whole manuscript. The review should not be presented by listing what has done by others.
The Conclusions should highlight the key findings from this review.
Provide a nice graphical abstract representing the overview of the MS with key highlights.
Author Response
Reviewer #3 - Comments
The paper “materials-1083926-A Review on the Design and Hydration Properties of Natural Polymer-Based Hydrogels” investigates the effect nature and role of water in the formation and stability of various types of hydrogels based on natural polymers and hybrid systems. Here are my questions and comments:
I think it is a parallel work with some new review papers that publish in recent months such as:
“Ali, A., & Ahmed, S. (2018). Recent advances in edible polymer-based hydrogels as a sustainable alternative to conventional polymers. Journal of agricultural and food chemistry, 66(27), 6940-6967.“
or
“Zia, K. M., Tabasum, S., Nasif, M., Sultan, N., Aslam, N., Noreen, A., & Zuber, M. (2017). A review on synthesis, properties and applications of natural polymer based carrageenan blends and composites. International journal of biological macromolecules, 96, 282-301.”
Or
“Bao, Z., Xian, C., Yuan, Q., Liu, G., & Wu, J. (2019). Natural Polymer‐Based Hydrogels with Enhanced Mechanical Performances: Preparation, Structure, and Property. Advanced healthcare materials, 8(17), 1900670.“
so, what makes this review different from the other and from the most recent ones?
Should be provided a comprehensive table between all of the nanofibers until now used.
Response: The authors appreciate the insightful comments provided by Reviewer #3. These comments have added to an improvement in the quality of this manuscript.
In comparison with the review articles outlined above, the key goal of the current contribution provides a coverage on the role of hydration effects and related phenomena such as the kinetics of hydration in hydrogel systems. By contrast, many of the reviews above describe the preparation, structure and properties, where the role of solvent on the structure-function relationship was not an area of emphasis.
In this review, several examples of novel natural polymer hydrogel systems with novel applications provide further insight on the structure-function relationship and the role of water in such systems. In the revised manuscript, the authors have included a Table of recent studies (past five years), as suggested by the reviewer (cf. Table 1). Therein we have reported on natural polymer hydrogel systems with novel applications that relate to the unique physicochemical properties of the materials (e.g. water uptake capacity, mechanical stability, fast release kinetics, rapid swelling-deswelling kinetics). These properties are largely determined by the role of water and its uptake kinetics relevant to hydration phenomena. Studies reporting on the structure (role/kinetics of water)-function (applications in biomedical, environment, agriculture, etc) are scarce, therefore the current studies contributes to addressing this knowledge gap.
In Section 6, the Natural Polymer-Based Hydrogels applications should be discussed. What is the future direction of the Natural Polymer-Based Hydrogels use?
Section of drawbacks and future could be increased quality of the manuscript.
A review paper not only should summarize recently published works, but also should contain critical and comprehensive discussions. Therefore, check writing for the whole manuscript. The review should not be presented by listing what has done by others.
The Conclusions should highlight the key findings from this review.
Provide a nice graphical abstract representing the overview of the MS with key highlights.
Response: A separate section on the Future Direction of Natural Polymer-Based hydrogels was added and relates to the use of various synthetic strategies (e.g. double network systems, nanocomposites, and hybrid systems) to address the limited mechanical properties of bio-based materials as the major drawback (See Section 7). Furthermore, future prospects of natural polymer-based hydrogels will focus on supramolecular systems [e.g. host/guest, (e.g. rotaxanes) and biomimetic interactions, rotaxanes, metal coordination, etc.] and 3D printing technology to offer platforms for new materials with advanced applications.
The revised manuscript provides a critical overview of the field of natural polymer hydrogels (especially starch-, chitosan-, cellulose-based systems) with a focus on the role of water and how it contributes to various applications in the environment, agriculture and the biomedical fields. The current study is anticipated to advance the field of hydrogel materials relevant to the design of natural polymers with desired properties and applications.
The conclusion section has been revised and a revised TOC graphic that further highlights the scope of the current work has been submitted to address the reviewer concerns.
The authors wish to acknowledge Reviewer #3 for the insightful and constructive comments, along with the opportunity to improve the quality of this manuscript submission.

Round 2
Reviewer 3 Report
In general, the manuscript “materials-1083926 A Review on the Design and Hydration Properties of Natural Polymer-Based Hydrogels” has met the requirements for acceptance. In my opinion, the authors took into account the comments of the reviewer.